



**Mixing layer height on the North China Plain and meteorological**
**evidence of serious air pollution in southern Hebei**
Xiaowan Zhu[1,3], Guiqian Tang[1,2*], Jianping Guo[4], Bo Hu[1], Tao Song[1], Lili Wang[1],
Jinyuan Xin[1], Wenkang Gao[1], Christoph Münkel[5], Klaus Schäfer[6], Xin Li[1,7], and
Yuesi Wang[1]
[1]State Key Laboratory of Atmospheric Boundary Layer Physics and Atmospheric
Chemistry (LAPC), Institute of Atmospheric Physics, Chinese Academy of Sciences,
Beijing 100029, China
[2]Center for Excellence in Regional Atmospheric Environment, Institute of Urban
Environment, Chinese Academy of Sciences, Xiamen 361021, China
[3]University of Chinese Academy of Sciences, Beijing 10049, China
[4]State Key Laboratory of Severe Weather & Key Laboratory of Atmospheric
Chemistry of CMA, Chinese Academy of Meteorological Sciences, Beijing 100081,
China
[5]Vaisala GmbH, 22607 Hamburg, Germany
[6]Atmospheric Science College, Chengdu University of Information Technology
(CUIT), Chengdu 610225, China
[7]Beijing Municipal Committee of China Association for Promoting Democracy,
Beijing 100035, China
*Correspondence to*: G. Tang (tgq@dq.cern.ac.cn)

**Abstract**
To investigate the spatiotemporal variability of regional mixing layer height (MLH)
on the North China Plain (NCP), multi-site and long-term observations of MLH with
ceilometers at three inland stations [e.g., Beijing (BJ), Shijiazhuang (SJZ), Tianjin
(TJ)] and one coastal site [e.g., Qinhuangdao (QHD)] were conducted from 16
October 2013 to 15 July 2015. The MLH at the inland stations on the NCP were
highest in summer and lowest in winter, while the MLH in the coastal area of Bohai
was lowest in summer and highest in spring. The regional MLH developed the earliest
in summer (at approximately 7:00 LT) and reached the highest growth rates (164.5 m
h$^{-1}$) at approximately 11:00 LT, while in winter, the regional MLH developed much
later (at approximately 9:00 LT), with the maximum growth rates (101.8 m h$^{-1}$)
occurring at 11:00 LT. As a typical site in southern Hebei, the annual mean of MLH at
SJZ was 464±183 m, which was 15.0 % and 21.9 % lower than that at the BJ
(594±183 m) and TJ (546±197 m) stations, respectively. Investigation of radiation and
wind shear at NCP revealed that the net radiation was almost consistent on a regional
scale, and the lower MLH in southern Hebei was mainly due to the 1.9-2.8-fold
higher intensity of wind shear on the northern NCP than in southern Hebei at an
altitude of 300-1700 m. Furthermore, the ventilation coefficient and the relative
humidity in southern Hebei were 1.1-2.1 times smaller and 13.2-22.1 % higher than



that on the northern NCP, respectively. As a result, severe haze pollution occurred
much more readily in southern Hebei and the annual means of near-ground $PM_{2.5}$
concentrations were almost 1.3 times higher than those of the northern areas. Due to
the unfavorable weather conditions, industrial capacity should be reduced in southern
Hebei, heavily polluting enterprises should be relocated and strong emission reduction
measures are required to improve the air quality.

## 1. Introduction

The convective boundary layer is the region where turbulence is fully developed.
The height of the interface where turbulence is discontinuous is usually referred to as
the mixing layer height (MLH) (Stull, 1988). The mixing layer is regarded as the link
between the near-surface and free atmosphere, and the MLH is one of the major
factors affecting atmospheric dissipation ability, which determines both the volume
into which ground-emitted pollutants can disperse, as well as the convective time
scales within the mixing layer (Seidel et al., 2010). In addition, continuous MLH
observations will be of great importance for the improvement of boundary layer
parameterization schemes and for the promotion of meteorological model accuracy.
Conventionally, the MLH is usually estimated from radiosonde profiles (Seidel et
al., 2010). Although meteorological radiosonde observations can provide high-quality
data, they are not suitable for continuous fine-resolution MLH retrievals due to their
high cost and limited observation intervals (Seibert et al., 2000). As the most
advanced method of MLH detection, remote sensing techniques based on the profile
measurements from ground-based instruments such as sodar, radar, or lidar that have
the unique vertically resolved observational capability are becoming increasingly
popular (Beyrich, 1997; Chen et al., 2001; He et al., 2005). Because sound waves can
be easily attenuated in the atmosphere, the vertical range of sodar is generally limited
to within 1000 m. However, the optical remote sensing techniques can provide higher
height ranges (at least several kilometers). The single-lens ceilometers developed by
Vaisala have been widely used in a variety of MLH studies (e.g., Emeis et al., 2004;
Emeis et al., 2009; Emeis et al., 2011; Eresmaa et al., 2006; Münkel et al., 2006;
Muñoz and Undurraga, 2010; Munkel and Rasanen, 2004; Schween et al., 2014;
Sokół et al., 2014; Tang et al., 2016; Tang et al., 2015b). Compared with other remote
sensing instruments, this type of lidar has special features favorable for long-term and
multi-station observations (Emeis et al., 2009; Wiegner et al., 2014; Tang et al., 2016),
including the low-power system, the eye-safe operation within a near infrared laser
band, and the low cost and ease of maintenance during any weather conditions
(excluding rainy, strong windy or sandstorm weather conditions) with only regular
window cleaning required (Emeis et al., 2004; Tang et al., 2016).
The North China Plain (NCP) region is the political, economic and cultural center
of China. With the rapid economic development, energy use has increased
substantially, resulting in frequent air pollution episodes (e.g., Guo et al., 2011; Li et
al., 2013; Liu et al., 2016; Tang et al., 2015a; Wang et al., 2014; Wang et al., 2013; Xu
et al., 2016; Zhang et al., 2014). The haze pollution has had an adverse impact on
human health (Tang et al., 2017a) and has aroused a great deal of concern (Tang et al.,
2009; Ji et al., 2012; Zhang et al., 2015). To achieve the integrated of development of





the Jing-Jin-Ji region, readjustment of the regional industrial structure and layout is
imperative. To this end, the industrial capacity of heavily polluting enterprises in the
areas with unfavorable weather conditions should be reduced, and these heavily
polluting enterprises should be removed to improve the air quality. For the remaining
enterprises, the industrial air pollutant emissions structure should be changed, and
strong emission reduction measures must be implemented. Although the government
has carried out some strategies for joint prevention and control, with the less
well-understood distributions of regional weather condition status on the NCP, how
and where to adjust the industrial structures on the NCP are questions in pressing need
of answers. As one of the key factors influencing the regional heavy haze pollution
(Tang et al., 2012; Quan et al., 2013; Hu et al., 2014; Tang et al., 2016; Zhu et al.,
2016; Tang et al., 2017b; Zhang et al., 2016), the MLH to some extent represents the
atmospheric environment capacity, and the regional distribution and variation of MLH
in the NCP can offer a scientific basis for regional industrial distribution readjustment,
which will be of great importance for regional haze management.
Nevertheless, due to the scarcity of MLH observations on the NCP, reliable and
explicit characteristics of MLH on the NCP remain unknown. Tang et al. (2016)
utilized the long-term observation data of MLH from ceilometers to analyze the
characteristics of MLH variations in Beijing (BJ) and verified the reliability of
ceilometers. The results demonstrated that MLH in BJ was high in spring and summer
and low in autumn and winter with two transition months in February and September.
A multi-station analysis of MLH in the NCP region was conducted in February 2014,
and the characteristics of high MLH at coastal stations and low MLH at southwest
piedmont stations were reported (Li et al., 2015). Miao et al. (2015) modeled the
seasonal variations of MLH on the NCP and discovered that the MLH was high in
spring due to the strong mechanical forcing and low in winter as a result of the strong
thermodynamic stability in the near-surface layer. The mountain-plain breeze and the
sea breeze circulations played an important role in the mixing layer process when the
background synoptic patterns were weak in summer and autumn (Tang et al., 2016).
However, the regional MLH simulation analysis is incomplete without verification
with long-term measured MLH data. To overcome previous studies' deficiencies, our
study first conducted a 22-months (from 16 October 2013 to 15 July 2015)
observation of MLH with ceilometers on the NCP. The observation stations included
three inland stations [e.g., BJ, Shijiazhuang (SJZ) and Tianjin (TJ)] and one coastal
site [e.g., Qinhuangdao (QHD)]. First, we will describe the spatial and temporal
distribution of MLH on the NCP. Subsequently, reasons for MLH differences on the
NCP will be explained in the discussion section. Finally, the weather conditions on the
NCP are described to provide a scientific basis for regional industrial structure
readjustment.

## 2  Data and methods

### 2.1 Sites

To study the regional MLH characteristics in the NCP region, observations with
ceilometers were conducted at the BJ, SJZ, TJ and QHD stations (Fig. 1 and Table 1).
The SJZ, TJ and QHD sites were set around Beijing in the east, southeast and




southwest direction, respectively. The BJ station was at the base of the Taihang and
Yanshan Mountains on the northern NCP. The MLH observation site was built in the
courtyard of the Institute of Atmospheric Physics, Chinese Academy of Sciences
(116.32° E, 39.90° N). SJZ was near the Taihang Mountain in southern Hebei; the
location was in the Hebei University of Economics (114.26° E, 38.03° N). The TJ site
was set in the courtyard of the Tianjin Meteorological Bureau, which was located
south of the urban area, with a geographic location of 117.20° E, 39.13° N. The QHD
station was an eastern coastal site of Bohai Bay, which was set up in the
Environmental Management College of China (119.57° E, 39.95° N) and the
surrounding areas are mostly residential buildings with no high structures.

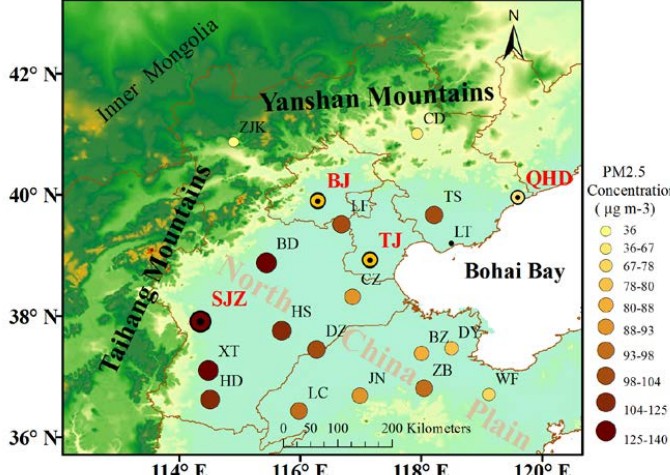

Fig. 1. Locations of the ceilometers observation sites (BJ, SJZ, TJ and QHD) are
marked with red and bold abbreviations; other PM$_{2.5}$ observation sites (ZJK, CD, LF,
TS, CZ, BD, HS, XT, HD, DZ, LC, JN, BZ, DY, ZB and WF) and the sounding
observation sites (BJ, LT and XT) are marked on the map with black abbreviations.
The size and color of the circular mark are representatives of the annual mean of
near-ground PM$_{2.5}$ concentration; the larger and darker the circle, the greater the
concentrations.



Table 1. Specific information of the observation sites on the NCP.

| Cityname | Abbreviation | Province or municipality | Longitude | Latitude |
|---|---|---|---|---|
| Beijing[a,b,c] | BJ | Beijing | 116.32° E | 39.90° N |
| Tianjin[a,b] | TJ | Tianjin | 117.20° E | 39.13° N |
| Shijiazhuang[a,b] | SJZ | Hebei | 114.26° E | 38.03° N |
| Langfang[a] | LF | Hebei | 116.70° E | 39.53° N |
| Tangshan[a] | TS | Hebei | 118.02° E | 39.68° N |
| Qinhuangdao[a,b] | QHD | Hebei | 119.57° E | 39.95° N |
| Zhangjiakou[a] | ZJK | Hebei | 114.92° E | 40.90° N |
| Chengde[a] | CD | Hebei | 117.89° E | 40.97° N |
| Laoting[b,c] | LT | Hebei | 118.90° E | 39.31° N |
| Cangzhou[a] | CZ | Hebei | 116.83° E | 38.33° N |
| Baoding[a] | BD | Hebei | 115.48° E | 38.85° N |
| Hengshui[a] | HS | Hebei | 115.72° E | 37.72° N |
| Xingtai[b,c] | XT | Hebei | 114.48° E | 37.05° N |
| Handan[a] | HD | Hebei | 114.47° E | 36.60° N |
| Dezhou[a] | DZ | Shandong | 116.29° E | 37.45° N |
| Liaocheng[a] | LC | Shandong | 115.97° E | 36.45° N |
| Jinan[a] | JN | Shandong | 116.98° E | 36.67° N |
| Binzhou[a] | BZ | Shandong | 118.02° E | 37.22° N |
| Dongying[a] | DY | Shandong | 118.49° E | 37.46° N |
| Zibo[a] | ZB | Shandong | 118.05° E | 36.78° N |
| Weifang[a] | WF | Shandong | 119.06° E | 36.68° N |

[a]Ceilometer observation sites.
[b]Near-ground $PM_{2.5}$ concentration sites.
[c]Radiosonde observation sites.

**2.2 Measurement of MLH**

The instrument used to measure the MLH at the four stations was an enhanced

single-lens ceilometer (Vaisala, Finland), which utilized the strobe laser lidar (laser
detection and range measurement) technique (910 nm) to measure the attenuated
backscattering coefficient profiles. As large differences existed in the aerosol
concentrations between the mixing layer and the free atmosphere, the MLH can be
determined from the vertical attenuated backscattering coefficient (β) gradient,
whereby a strong, sudden change in the negative gradient (-dβ/dx) can indicate the
MLH. In the present study, the Vaisala software product BL-VIEW was utilized to
calculate the MLH by determining the location of the maximum -dβ/dx in the
attenuated backscattering coefficient. To strengthen the echo signals and reduce the
detection noise, spatial and temporal averaging should be conducted before the
gradient method is used to calculate the MLH. The BL-VIEW software was utilized
with temporal smoothing of 1200 s and vertical distance smoothing of 240 m. The
instrument installed at the BJ station was a CL31 ceilometer and the CL51





ceilometers were used at the SJZ, TJ and QHD stations. Some of the properties of
these two instruments are listed in Table 2, and basic technical descriptions can be
found in Münkel et al. (2007) and Tang et al. (2015).
To ensure the consistency of the MLH measured with the two different versions of
ceilometers, before we set up the ceilometers at different stations, we made a
comparison of the MLH observed by CL31and CL51 at BJ from October 1 to October
8, 2013 (Fig. S1). The MLH observed by CL 31 was highly correlated with those
observed by each of the three CL51 ceilometers, with relative correlation coefficients
(R) of 0.92, 0.86 and 0.92. Therefore, the impact of version discrepancy on MLH
measurement can be neglected.
Table 2. Instrument properties of CL31 and CL51

| Parameter | CL31 | CL51 |
|---|---|---|
| Detection range (km) | 7.7 | 13.0 |
| Wavelength (nm) | 910 | 910 |
| Report period (s) | 2-120 | 6-120 |
| Report accuracy | 5m | 10m |
| Peak power (W) | 310 | 310 |

**2.3 Other data**
The hourly data of relative humidity (RH), temperature (T), near-ground wind
speed (WS) and direction at the BJ, SJZ, TJ and QHD stations were obtained from
China                    Meteorological                    Administration
(http://www.weather.com.cn/weather/101010100.shtml). Hourly net (0.2-100 μ m)
radiation data at the BJ, TJ and SJZ sites were observed using a net radiometer (NR
Lite2, Kipp & Zonen, Netherlands), detailed information is included in Hu et al.,
(2012). Because the SJZ and QHD stations are missing radio sounding data, sounding
data from the XT and LT stations were used instead. Sounding data of WS and
direction at the BJ, XT and LT stations were provided by the upgraded radiosonde
network of China, where the GTS1 digital electronic radiosonde was required to be
operationally launched twice per day at 08:00 LT and 20:00 LT by the China
Meteorological Administration (Guo et al., 2016).
The near-ground $PM_{2.5}$ and $PM_{10}$ concentrations at the 20 observation sites from
December 2013 to November 2014 were provided by the Ministry of Environmental
Protection (http://www.zhb.gov.cn/) with a time resolution of 1 h. Details for the
near-ground $PM_{2.5}$ and $PM_{10}$ observation sites are listed in Table 1 and Fig. 1.
**3   Results**
**3.1 Frequency distribution of regional MLH**
Continuous operation of the ceilometers since October 2013 has provided 22
months of data, and for the purpose of analyzing of the MLH variability in the NCP
region, the hourly averages of MLH for a whole year (from December 2013 to
November 2014) at the BJ, SJZ, TJ and QHD stations were utilized in the following
study. Hourly means of MLH under rainy, sandstorm and windy conditions were
removed (Muñoz and Undurraga, 2010; Tang et al., 2016; van der Kamp and
McKendry, 2010), resulting in data availability of 81, 89, 83 and 77 % at the BJ, SJZ,
TJ and QHD stations, respectively. The frequency distribution of daily maximum





MLH is shown in Fig. 2. In this study, March, April and May are defined as spring;
June, July and August are defined as summer; September, October and November are
defined as autumn; and December, January and February are defined as winter.
The daily maximum MLH at the BJ, SJZ and TJ stations reached 2400 m, and the
high daily maximum values mostly occurred in spring and summer, while the low
values occurred in autumn and winter and were as low as 200 m. The maximum MLH
values at the BJ, SJZ and TJ stations were mainly distributed between 600 and 1800
m, 400 and 1600 m and 800 and 1800 m, respectively, and they accounted for 74.2,
72.0 and 67.0 % of the total samples, respectively. Notably, the daily maximum MLH
at SJZ was lower in spring, autumn and winter in comparison with those at the BJ and
TJ stations. Values below 600 m at the SJZ station occurred primarily in autumn and
winter; the most frequent daily maximum MLH was in the range from 1000 to 1200
m, which was 200-600 m lower than that at the TJ station. This pattern demonstrated a
weaker atmospheric diffusion capability at SJZ in spring, autumn and winter than at
the northern stations.
The frequency distribution of the daily maximum MLH at the coastal station was
different. The daily maximum MLH at QHD was mainly distributed between 800 and
1800 m with relatively uniform seasonal distributions (Fig. 2d). Values lower than 600
m mainly occurred in summer, which was probably influenced by the frequent
occurrence of a thermal internal boundary layer in summer (van der Kamp and
McKendry, 2010).

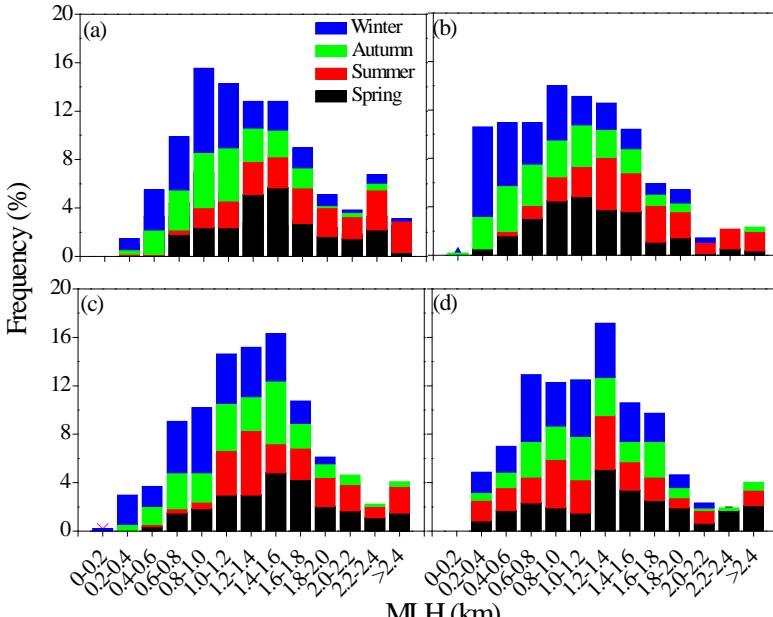

Fig. 2. Frequency distribution of daily maximum MLH at the (a) BJ, (b) SJZ, (c) TJ
and (d) QHD stations from December 2013 to November 2014.
**3.2 Spatiotemporal variation of regional MLH**
**3.2.1 Seasonal variation**





Monthly variations of MLH at the BJ, SJZ, TJ and QHD stations are shown in Fig.
3. The monthly means of the regional MLH ranged between 300 and 750 m; the
maximum and minimum MLH occurred in June 2014 at the BJ station and in January
2014 at the SJZ station, with values of 741 and 308 m, respectively. Most of the
monthly averages were between 400 and 700 m, which accounted for 81.3 % of the
total samples.
The MLH at the BJ, SJZ and TJ stations showed obvious seasonal variations with
high values in spring and summer and low values in autumn and winter. Seasonal
means   of   MLH   at   the   three   stations   followed   the   same   order:
summer>spring>autumn>winter, with maximum values of 722±169, 623 ±161 and
655±165 m in summer, respectively, and minimum values of 493±131, 347±153 and
436±178 m in winter, respectively (Table S1). Obvious annual changes of the MLH
with large amplitude at the BJ, SJZ and TJ stations implied that MLH is influenced by
seasonal changes of solar radiation, and in summer, the intense solar radiation favors
the development of MLH (Stull, 1988).
Nevertheless, the seasonal variation of MLH at the coastal site of Bohai was
different from that at the inland stations. The MLH at QHD exhibited a decreasing
trend from spring to summer and increasing trend from autumn to winter, and the
maximum seasonal mean at QHD was 498±217 m in spring and the minimum was
447±153 m in summer. Moreover, the MLH in spring and summer at QHD was much
lower than at other stations. Similar to our analysis of frequency distributions of daily
maximum MLH in Section 3.1, the lower MLH at QHD in spring and summer mainly
resulted from the frequent occurrence of sea breeze (Fig. 5). Under the influence of
the abrupt change of aerodynamic roughness and temperature between the land and
sea surfaces, a thermal internal boundary layer will occur frequently in the coastal
areas, which will decrease the average MLH to some extent. This impact of sea breeze
on the coastal boundary layer was consistent with previous studies (Zhang et al., 2013;
Tu et al., 2012), which demonstrated that ceilometers can properly retrieve the coastal
MLH as well.

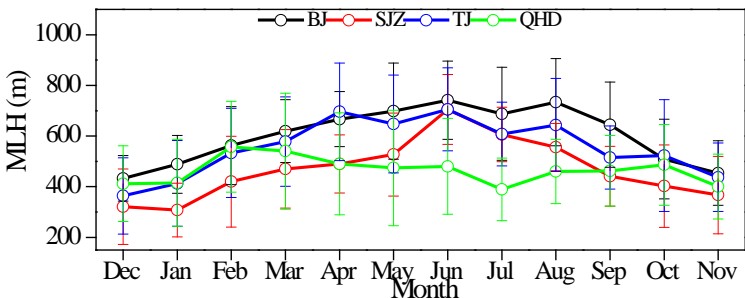


Fig. 3 Monthly variations of MLH at the BJ, SJZ, TJ and QHD stations from
December 2013 to November 2014.
Annual averages of MLH at the BJ, SJZ, TJ and QHD stations were also calculated,
and the values were 594±183, 464±183, 546±197 and 465±175 m, respectively. The
MLH at SJZ was approximately 21.9, 15.0 and 0.2 % lower than at the BJ, TJ and



QHD stations, respectively, which revealed a more stable atmospheric stratification
and weaker atmospheric environment capability in southern Hebei.

**3.2.2 Diurnal variations**

Seasonal variations of diurnal MLH change patterns were investigated to reveal the
24 h evolution characteristics of the regional MLH on the NCP. As shown in Fig. 4,
diurnal variations of regional MLH in different seasons all had single peak patterns.
With sunrise and increased solar radiation, MLH at the four stations started to develop
and peaked in the early afternoon. After sunset, turbulence in the MLH decayed
quickly, and the mixing layer underwent a transition to the nocturnal stable layer (less
than 400 m). The averaged annual daily ranges of MLH at the BJ, SJZ, TJ and QHD
stations were 782, 699, 914 and 790 m, respectively, and the averaged annual daily
range of MLH at SJZ was 100-200 m smaller than at other stations. When we referred
to the diurnal variations of regional MLH in different seasons, we found that the
lower annual daily range at the SJZ station was associated with its lower values of
daytime MLH in spring, autumn and winter (Figs. 4a, 4c and 4d).
Average growth rates for the four stations demonstrated that the growth rates of the
regional MLH varied by season. The MLH developed earliest in summer (at
approximately 7:00 LT) and reached the highest growth rates (164.5 m h$^{-1}$) at
approximately 11:00 LT, and the time when MLH started to develop was found to be
1 hour later (at approximately 8:00 LT) in spring and autumn than in summer.
Furthermore, the MLH developed the latest (at approximately 9:00 LT) and slowest in
winter, with the maximum growth rate (101.8 m h$^{-1}$) occurring at approximately 11:00
LT.

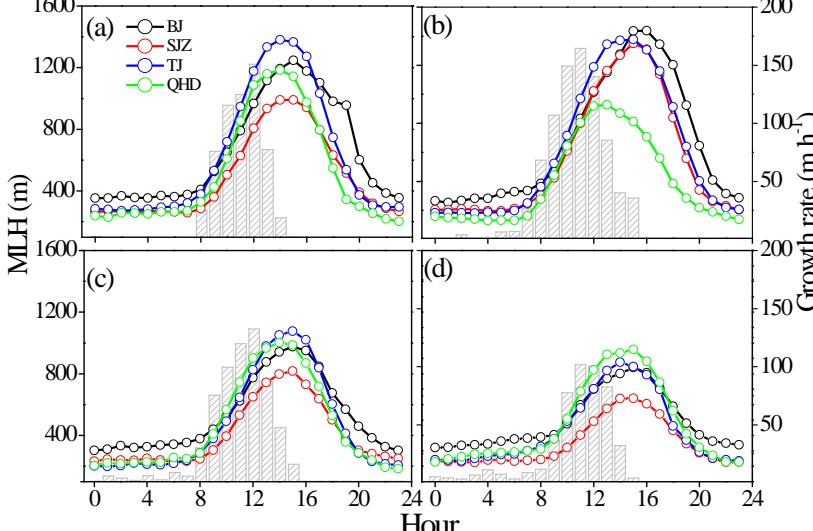

Fig. 4. Diurnal variations of MLH at the BJ, SJZ, TJ and QHD stations in (a) spring,
(b) summer, (c) autumn and (d) winter seasons are indicated by lines and scatters. The
averaged growth rates at the four sites are depicted with gray columns for each season
to represent the regional MLH growth velocity, and only positive values are shown in





the figure.
Comparison of the MLH peaking time between the four stations showed that the
maximum MLH at the TJ and QHD stations arrived earlier than at the BJ and SJZ
stations in spring and summer (Figs. 4a and 4b). However, in autumn and winter, such
a characteristic was not evident (Figs.4c and 4d).
As shown in Fig. 5, under the influence of the Siberian High and the geographic
location effect, northerly and northwesterly winds prevailed in autumn and winter at
the four stations. In spring and summer, the northward lift and westward intrusion of a
subtropical high causes the weak southerly wind to arrive and dominate in the NCP
region. Without a large- or medium-scale weather system passing through, the sea
breeze will play a role in the coastal area. Although the TJ station was supposed to be
an inland site, it was still affected by the sea breeze to some extent. Due to the
shoreline orientation and regional topography differences between TJ and QHD (Fig.
1), when a sea breeze occurred, easterly wind prevailed at the former station and
easterly, and south-southwesterly wind blew at the latter station in spring and summer
(Figs.5c and 5d). Statistical results revealed that from March 2014 to August 2014, the
frequency of sea breeze occurrence at the TJ and QHD stations could reach 53.8 and
92.4 %, respectively, and the sea breeze usually started at midday (approximately
11:00 LT).
Generally, the vertical development of the mixing layer is heavily reliant on the
vertical turbulence, but when sea breeze is present, cool air advection from the sea
breeze circulation will suppress this vertical mixing intensity (Puygrenier et al., 2005).
The co-existence of vertical turbulence and advection caused the MLH to decrease
and peak earlier. Meanwhile, the local mixing layer will be replaced by the thermal
internal boundary layer (Tomasi et al., 2011). As a result, the earlier peaking time of
MLH in spring and summer could be attributed to the sea breeze effect. The MLH
peaking time at the TJ station was approximately 1-2 hours later than at the QHD
station, which indicated that such a sea breeze impact will weaken with distance from
the coast (Huang et al., 2016).













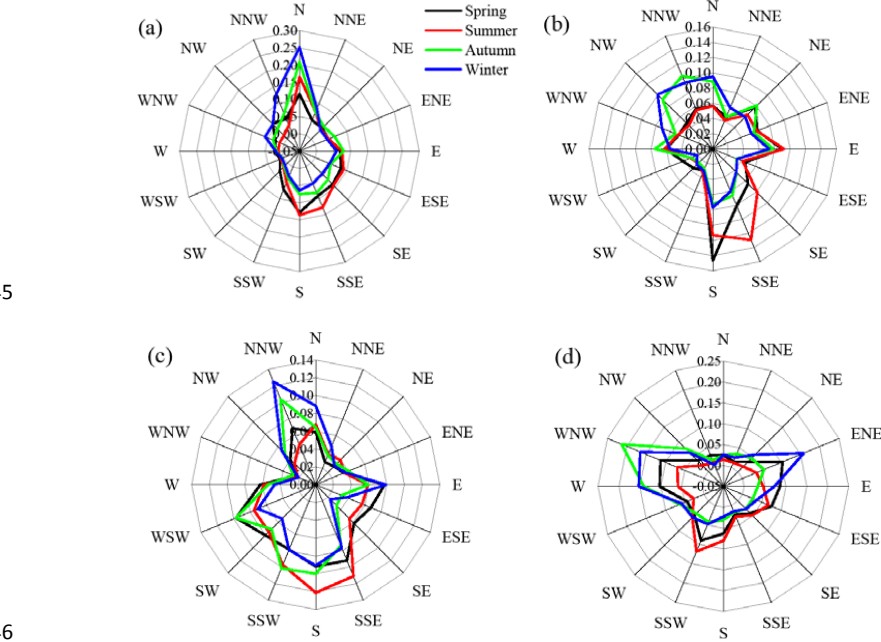



Fig. 5 Frequency of wind direction at the (a) BJ, (b) SJZ, (c) TJ and (d) QHD stations in different seasons.

Therefore, according to the analysis above in Sections 3.1 to 3.2, an obvious phenomenon can be observed in the MLH distribution in the NCP region: the MLH was lower in southern Hebei than on the northern NCP in spring, autumn and winter but was almost equal to the northern areas in summer.

**4. Discussion**

**4.1 Reasons for low MLH in southern Hebei**

Turbulent energy was mainly responsible for the MLH development, and the generation of turbulent energy was highly correlated with the buoyancy flux (mainly heat and moisture fluxes) produced by net radiation and the momentum flux caused by wind shear (*Stull*, 1988). We first compared the net radiation among the BJ, SJZ and TJ observation sites. As shown in Fig. 6, the seasonal net radiation variations were almost consistent among the three stations, and they were high in spring and summer and low in autumn and winter, with annual averages of 5.4, 6.0 and 4.8 W $m^{-2}$, respectively. The comparable net radiation values at the BJ and SJZ stations indicated that the buoyancy flux was unable to explain the MLH differences between the northern NCP and southern Hebei.





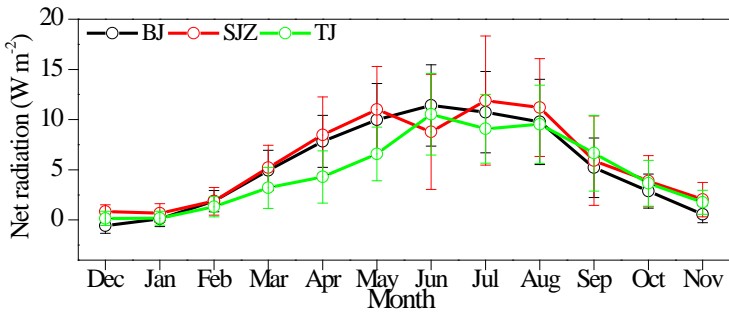


Fig. 6 Monthly variations in net radiation at the BJ, SJZ and TJ sites.

Wind shear was defined and calculated according to Eq. (1):

$$\text{wind shear} = \sqrt{\left(\frac{du}{dz}\right)^2 + \left(\frac{dv}{dz}\right)^2} \qquad (1)$$

where $dz$ is the height difference between two layers at which the vertical wind shear
is estimated and $du$ and $dv$ are the differences in zonal and meridional directions in
the two different layers (Hyun et al., 2005). Considering the geographic locations (Fig.
1), the lack of sounding data at the SJZ station was addressed by replacement with
sounding data from another southern Hebei station (e.g., the XT station); meanwhile,
sounding data from another coastal site (e.g., the LT station) were used instead of
from the QHD station. Observations were conducted at 8:00 LT and 20:00 LT each
day from December 2013 to November 2014, and the wind shear was averaged every
100 m for each sounding profile.
When we analyzed the seasonal means of wind shears between southern Hebei (XT)
and the northern NCP (BJ), some distinct features were observed, as shown in Fig. 7.
Considering that the regional MLH at 08:00 LT and 20:00 LT was mostly below 300
m (Fig. 4), wind shears in southern Hebei were lower than those in the northern NCP
below 300 m but were nearly consistent at the altitude of 300 m both at 08:00 LT and
20:00 LT during the whole year. However, above 300 m at 08:00 LT, wind shears at
XT were significantly different from those at BJ again at the altitude of 300-1700 m
and, on average, approximately 2.8, 2.5 and 1.9 times smaller than at the BJ stations
in spring, autumn and winter, respectively (Figs. 7a, 7c and 7d). The largest
discrepancies reached 3.4, 4.3, and 4.5 m s$^{-1}$ km$^{-1}$ in spring, autumn and winter,
respectively, and were at the altitude between 500 and 700 m. In summer, the
averaged differences narrowed down to only 1.2-fold above 300 m (Fig. 7b).
Compared to wind shears at 20:00 LT above 300 m in spring, autumn and winter,
mechanical forces were clearly enhanced in BJ at the height of 300-1700 m during the
whole night and the turbulent energy was restored in the residual layer. With the
increase of solar radiation in the morning, the MLH developed and broke through the
residual layer. At this time, the combination of buoyancy and wind shear forces will
contribute to a higher MLH at BJ during daytime. Furthermore, the larger wind shears
below 300 m during night time at the BJ station could partly explain the higher
nocturnal boundary layer on the northern NCP (Fig. 4).



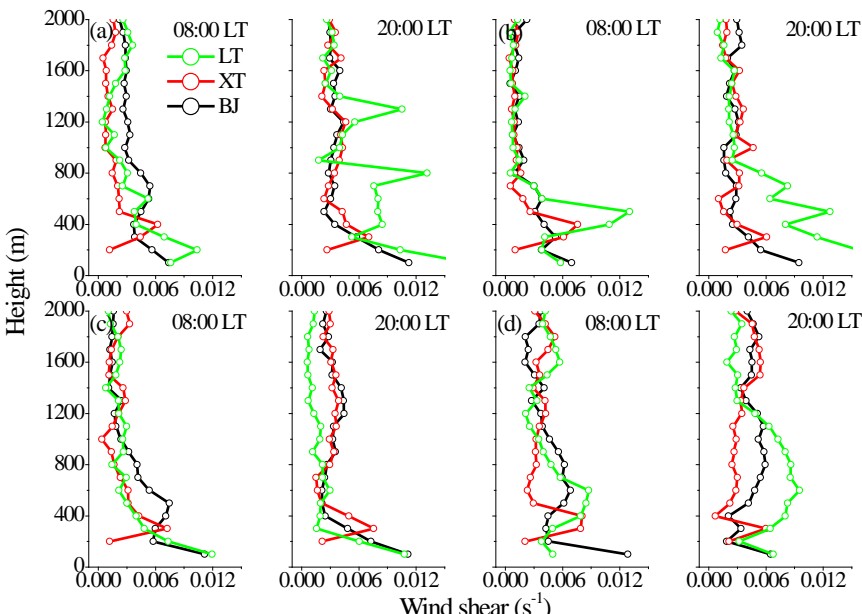


Fig.7 Vertical profiles of wind shear at the BJ, XT and LT stations in (a) spring, (b)
summer, (c) autumn and (d) winter.

The lower MLH in southern Hebei was the result of the lessened mechanical
forcing due to wind shear at night than occurred in the northern areas in spring,
autumn and winter. This pattern could be attributed to the influence of the active
fronts passing by under the impact of the Siberian High, and usually, this front system
does not reach southern Hebei. In summer, due to the influence of the subtropical high
on the NCP and the relatively greater solar radiation, the lessened effects of the front
system and strong turbulent exchange will lead to less wind shear contrast in the
vertical direction between southern Hebei and the northern NCP.

**4.2 Meteorological evidence of serious air pollution in southern Hebei**

When we analyzed the near-ground $PM_{2.5}$ and $PM_{10}$ concentrations distribution on
the NCP from December 2013 to November 2014, a unique phenomenon was found
and shown in Fig. 1 and Fig. S3. The annual means of near-ground $PM_{2.5}$
concentration in southern Hebei (SJZ, XT, HS, HD and DZ) was 124.1μg m$^{-3}$ (218.8
μg m$^{-3}$ for the $PM_{10}$ concentrations), while in the northern areas (BJ, TJ, LF and TS),
it was 94.9 μg m$^{-3}$ (145.5 μg m$^{-3}$ for the $PM_{10}$ concentrations), and the difference in
near-ground $PM_{2.5}$ concentration between these two areas can be as high as 1.3-fold
(1.5-fold for the $PM_{10}$ concentrations). Considering the low MLH in southern Hebei,
heavy pollution in southern Hebei may be related with weaker weather conditions,
and some other meteorological factors may play a part.

Previous studies revealed that the most significant meteorological factors for
regional heavy haze formation on the NCP were RH and MLH (Tang et al., 2016; Zhu
et al., 2016). However, due to the lack of wind profiles, Tang et al. (2015) utilized the
near-surface WS to estimate the ventilation coefficients ($V_c$), and the result was not





sufficiently precise and could not portray the regional pollution dissipation ability
accurately. In this study, we utilized wind sounding data to enable an exact evaluation
of the regional pollutant dissipation ability. Furthermore, temperature is the main
factor in new particle formation, and RH determines the growth rates of particles,
which are the most influential meteorological factors for particle formation. As a
consequence, in the next section, we will separately analyze the regional particle
formation and dissipation ability, each from a meteorological point of view.
**4.2.1 Meteorological factors for particle formation**
Monthly variations of T and RH are shown in Fig. 8. The T in the southern Hebei
was similar to that on the northern NCP in variation pattern and quantity but was
approximately 19.3 % higher than at the coastal site (Fig. 8a). Under the same
temperature conditions, the new particle formation ability will be the same between
these two areas. However, differences existed in RH between southern Hebei and the
northern NCP. The RH at the SJZ station was always higher than at the BJ and TJ
stations but was slightly lower than at the QHD station through the year (Fig. 8b). The
annual averages of RH at the BJ, SJZ, TJ and QHD sites were 51.2, 65.7, 57.0 and
68.6 %, respectively, and the RH at SJZ was 22.1 and 13.2 % higher than at the BJ
and TJ sites, respectively (Table S2). As RH is also a key factor for haze development
and determines the particle growth rate through hygroscopic growth and secondary
formations (Zhao et al., 2013; Fu et al., 2014), even though the new particle formation
conditions were the same between these two areas, particles can grow larger under
high RH, leading to heavier pollution in southern Hebei.

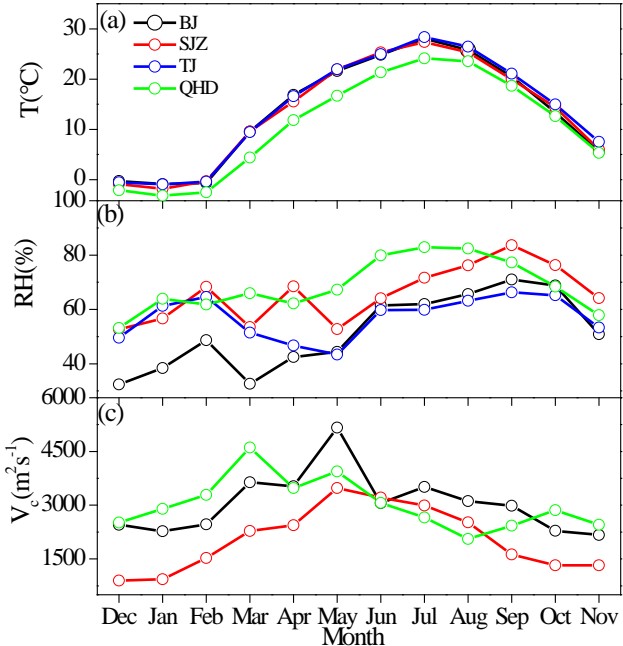


Fig. 8 Seasonal variations of (a) T, (b) RH and (c) $V_c$ at the BJ, SJZ, TJ and QHD
stations from December 2013 to November 2014.





**4.2.2 Meteorological factors for particle dissipation**

As MLH and WS can represent the atmospheric dissipation capability in the
vertical and horizontal directions, respectively, in addition to the MLH, we also
analyzed the WS variations on the NCP. Similar to our analysis in Section 4.1, as SJZ
and QHD had no sounding data and due to the close geographic proximity among SJZ
and XT as well as LT and QHD, sounding data from the XT and LT stations were used
instead of the data at SJZ and QHD, respectively. The WS profiles were averaged
every 100 m at each stations and are depicted in Fig. S2. Except for summer, the WS
in southern Hebei was far less than that on the northern NCP and coastal areas both at
08:00 LT and 20:00 LT in spring, autumn and winter (Fig. S2a, S2c and S2d) but was
nearly consistent in summer (Fig. S2b). This finding indicated a weaker horizontal
diffusion capability in southern Hebei than on the northern NCP and at the coastal
sites.
The ventilation coefficient is an important factor in pollutant dissipation and air
quality studies; it accounts for the vertical dispersion and advection of pollutants.
With larger $V_c$, strong dissipation ability follows. The $V_c$ is defined as the product of
MLH to the wind transport ($U_T$) and is shown in Eq. (2).
$$V_c = MLH \times U_T \tag{2}$$
When we utilized the wind profiles in Fig. S2 with equal spacing in the vertical
direction, $U_T$ could be regarded as the mean wind transport, i.e., $U_T = \frac{1}{n}\sum_{i=1}^{n} U_i$ and $U_i$
is the wind observed at each level and n is the number of levels within the mixing
layer (Nair et al., 2007). As the profiles of WS for each station were almost the same
in the morning and at night (Fig. S2), it was considered reasonable to regard the
sounding data of WS as a climatological constant, and the $V_c$ within the mixing layer
could then be calculated. Considering the monthly averaged MLH at the BJ, SJZ and
QHD stations, the monthly $V_c$ is depicted in Fig. 8c. $V_c$ at the southern Hebei was
always lower than the northern NCP during the whole study period. The seasonal
means of $V_c$ at the BJ, SJZ and QHD stations in spring, summer, autumn and winter
were 4112.0, 2733.3 and 4008.5; 3227.5, 2908.8 and 2593.7; 2481.4, 1421.9 and
2581.7; and 2397.2, 1117.7 and 2900.0 $m^2\ s^{-1}$, respectively. It was clear that the SJZ
station usually had the lowest $V_c$, and the annual averaged $V_c$ at SJZ was almost 1.5
and 1.5 times smaller than the BJ and QHD stations, respectively (Table S2). As a
result, the particle dissipation capability in southern Hebei was much weaker than in
the northern and coastal areas.
Therefore, due to the lower atmospheric environment capability, the weaker
dissipation ability and stronger particle formation ability, the particles were more
easily accumulated and severe haze occurred frequently in southern Hebei. This
finding indicated that the industrial structure of southern Hebei is in need of
readjustment.
**5. Conclusions**
To gain new insight into the spatiotemporal variation of the regional MLH, the
present study conducted a simultaneous observation with ceilometers at three inland
stations (e.g., BJ, SJZ, and TJ) and one coastal site (e.g., QHD) to obtain high spatial





and temporal resolution MLH data. The experiment period lasted for 22-months from
October 16, 2013, to July 15, 2015, and one year's data (e.g., from December 2013 to
November 2014) were utilized for further study. Conclusions were drawn as follows.
The MLH in the inland areas of the NCP was high in spring and summer and low in
autumn and winter. Under the effects of sea breeze and a thermal internal boundary
layer, the seasonal variation of the MLH in the coastal area of Bohai was different
from that of the inland stations, and the lowest MLH was occurred in summer. The
MLH peaked earlier at the coastal site in spring and summer than at the inland
stations, and this effect weakened with distance from the coast. This effect of sea
breeze on coastal MLH was consistent with previous studies, which demonstrated that
not only can the mainland MLH be retrieved from ceilometers, but the coastal MLH
can be observed with ceilometers.
The MLH in southern Hebei was lower than that on the northern NCP, especially in
spring, autumn and winter. As there was little radiation difference between these two
areas, the lower MLH in the southern Hebei mainly resulted from the stronger
intensity of wind shears on the northern NCP than in southern Hebei at an altitude of
300-1700 m in residual layers. In summer, the wind shear difference lessened, and the
MLHs between the southern and northern areas were nearly consistent.
From a meteorological point of view, the weaker atmospheric environment
capability combined with the weaker pollutant dissipation ability and the stronger
pollutant formation ability will cause severe haze to occur easily in southern Hebei,
and the industrial layout in southern Hebei is in need of restructuring. Heavily
polluting enterprises should be relocated to locations with better weather conditions
(e.g., certain northern areas and coastal areas), and strong emission reduction
measures should be implemented in the remaining industrial enterprises to improve
air quality.
Overall, the present study is the first to conduct a long-term observation of the
MLH with high spatial resolution on a regional scale. The observation results will be
of great importance for model parameterization scheme promotion and provide basic
information for the distribution of weather conditions in the NCP region. The
deficiency of this study is that we took no account of the transport effect on $PM_{2.5}$
concentrations. Because pollutants are usually transported from south to north in the
NCP region during haze episodes (Zhu et al., 2016; Tang et al., 2015), the pollutant
transport has a greater impact on northern areas and had less of an influence on the
results of this analysis. The absence of sounding data at noon is another shortcoming,
and we plan to conduct daytime observations in future experiments. Nevertheless, our
study can provide reasonable and scientific suggestions for industrial layout and air
pollution emissions reduction measures for the NCP region, which will be of great
importance for achieving the integrated development goals.

Acknowledgments
This work was supported by the CAS Strategic Priority Research Program (Grant
no.XDB05020000), the National Natural Science Foundation of China (Grant
no.41230642) and the National Earth System Science Data Sharing Infrastructure,



National Science & Technology Infrastructure of China

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
