# Peer review of "Mixing layer height on the North China Plain and meteorological"

_Atmospheric Chemistry and Physics, 2017_

## Referee Comment (RC1) · Anonymous Referee #1 · 10 Jul 2017

This paper characterizes mixing layer height (MLH) over the major cities in the North China Plain based on the two-year surface observations. The relationship between MLH and regional air pollution is explored using concurrent PM, MLH, surface radiation, and meteorological parameters in the same cities. Overall, the paper is well written and the finding about the low MLH in southern Hebei is valuable to develop an efficient air pollution mitigation strategy in North China. I suggest the paper should be accepted by ACP after the authors address my comments below.

1) It is not clear what is the difference between the MLH discussed here and the traditional defined planetary boundary layer height (PBLH). It would be interesting to see if

[Figure]

the MLH obtained from surface can be intercompared with PBLH from soundings like Guo J. et al. (2016).

2) L266, the authors attribute the lower summertime MLH in QHD to the higher frequency of sea breeze. However, the underlying physical mechanism is not fully explained. Intuitively, the active sea breezes should come with more unstable atmosphere over the land. Figure 5 about prevailing wind directions in different seasons is referred, but it is still unclear to me how this figure supports the hypothesis above. Some detailed discussions are needed to better describe the formation and characteristics of the sea breeze in the coastal regions.

3) L372, to overcome the lack of radio sounding in SJZ, how about directly using the reanalysis data? The quality of reanalysis can be evaluated by radiosound at XT.

4) Section 4.1, could absorbing aerosols be another factor to explain the reason of the low MLH in southern Hebei? Observations have revealed that the ambient aerosols can become highly absorptive in the urban conditions in China [Peng J. et al., 2016, PNAS]. The strong solar absorption near the top of PBL can increase the atmospheric stability and convective inhibition energy [Wang Y. et al., 2013, AE; Li Z. et al., 2016, Rev. Geos.]. Those possible influences from the feedback of air pollution should be discussed and quantified if possible.

5) L437, what makes the RH at SJZ is higher than that in BJ and TJ? SJZ is more inland than those two cities.

6) L432-445, some basics of new particle formation in urban condition should be thoroughly reviewed. Please refer to Zhang, R. 2010, Science and 2015, Rev. Chem.

7) Fig. 8. Define Vc in the figure caption.
* * *

---

## Short Comment (SC1) · 23 Aug 2017

The climatology of MLH at four sites over NCP were investigated using long-term measurements. However, lots of statements in the manuscript and part of conclusions were not well supported. Thus, a major revision is suggested.

1) LINE 214-215, the definitions of rainy, sandstorm and windy conditions should be given. 2) LINE 317-318, "the TJ station was supposed to be an inland site", the TJ site is quite close to the Bohai sea, which should be considered as a coastal station. 3) LINE 319-324, the definition of sea-breeze used in this study should be given. The sea-breeze cannot be identified merely by the near-surface wind speed and direction.

[Figure]

How to identify the sea-breeze from background wind? How to calculate the occurrence frequency of sea-breeze at TJ and QHD? 4) LINE 326-335, more evidences should be given to support the statement that the movement of sea-breeze suppress the MLH at QHD site in summer. The TJ site also locates in the coastal regions, why the diurnal patterns and seasonal variations of MLH are quite different? 5) LINE 362-364, the buoyancy fluxes are determined by the surface sensible heat fluxes, not the net radiations. The statements here are inaccurate. 6) LINE 371-375, before using the sounding data of XT as a replacement of SJZ, the data consistency must be examined and presented, since there are ~90 km between these two sites. At least, the general characteristics of MLH at SJZ at 08:00 and 20:00 LT should be well reflected by the sounding data at XT. The data consistency also should be check between the LT site and QHD site. 7) As shown in Fig. 7, the profiles at XT are almost the same in different season and different moment, which is different from the profiles of other sites. The prevailing wind speed and direction are different in different season, why the profiles are almost the same? The error-bar of the profiles should also be given. In spring and summer, at 20:00 LT there are lots of fluctuations in the profiles at LT, why? Do the terrains play a role in the profiles in different regions? 8) LINE 390-392, the authors merely presented the profiles at 20:00 LT, which cannot support the statement "during the whole night". More evidences should be given. 9) LINE 404-405, please give evidences to support the statement "the front usually does not reach southern Hebei". 10) LINE 406-408, please give evidences to support the statement "the lessened effects of the front system and strong turbulent exchange will lead to less wind shear contrast in the vertical direction between southern Hebei and the northern NCP." 11) LINE 410-419, the authors attribute the high PM concentration in SJZ to the low MLH. It is inaccurate, the different anthropogenic emissions of pollutants in SJZ and BJ should be considered. 12) LINE 420-422, although the RH can affect the visibility, it cannot significantly affect the aerosol concentration. Is there any direct physical connections between the high RH conditions and high aerosol concentration? 13) LINE 426-427, "temperature is the main factor in new particle formation," any evidences to

support this statement in NCP. 14) LINE 437-440, the RH in SJZ is higher than that in TJ (closer to sea), why? 15) Section 4.2.1, the authors attribute the higher PM in SJZ to new particle formation, which is quite complex and cannot be understood merely by the surface temperature and RH. And the direct emissions of pollutants should be considered. 16) LINE 470-473, "it was considered reasonable to regard the sounding data of WS as a climatological constant", during a day, the WS within ML would change due to the momentum exchanges between the ML and free troposphere. The WS cannot be considered as a constant. As illustrated in Fig. S2, there are differences in profiles at 08:00 and 20:00 LT. The error-bar of wind speed should be given.
* * *

---

## Referee Comment (RC2) · Anonymous Referee #3 · 27 Oct 2017

This study reveals the spatial variation of mixing layer height (MLH) over northern China plain (NCP) based on a two-year measurement at four primary cities with different geographical location across NCP. The authors attribute the different spatial pattern of MLH between southern Hebei and northern NCP to the distinct wind shear features between the two interested regions. The analysis on the long-term measurement of MLH in this study provides a meaningfully insight on the climatological features of boundary layer condition during the haze episodes over NCP. Also, the discussions about the associations of MLH and other meteorological factors with the near-ground particle pollution are sufficiently presented in this work. However, the following concerns should be addressed before publication.

Specific comments

1. Considering the possible strong aerosol-radiation interaction because of the heavily pollution, the surface net radiation is supposed to be lower over the regions with more heavily pollution because of the strong scattering and/or absorbing of solar radiation by aerosols. However, in this study, though the near-ground PM2.5 concentration over southern Hebei is 1.3 times higher than that of north China plain (NCP), there is no significant difference in the net radiation at Shijiazhuang (SJZ) located southern Hebei from at Beijing (BJ) located over NCP. One probable reason is because the aerosol optical depth (AOD) over the two sites are comparable, leading to comparable capacity reducing solar radiation. The authors may check the AOD data to obtain a convinced explanation for why the net radiation is spatial consistent, given the presence of aerosol-radiation interaction.

2. In addition to the difference in mixing layer height (MLH), how likely does the spatial variation in pollutant emissions contribute to the difference in the near-ground PM pollution between SJZ and BJ?

3. Th authors attribute the spatial difference in wind shear over NCP during winter to the influence of front passing associated with the Siberian High (lines 403-405). Is the front also the dominant control of the relative humidity over NCP during winter? Is there any other reason leading to the discrepancy in relative humidity between the two regions in question?

4. Given that both Tianjin (TJ) and Qinhuangdao (QHD) are located at coastal region and suffering highly frequent sea breezes during summer (Fig. 5), why the MLH of TJ is much higher than the case in QHD, since the relatively low MLH in QHD is attributed by the authors to the intensive occurrence of sea breeze during summer (lines 265-266)?

Technical comments

1. Fig. 7: the unit for the wind shear should be m s-1 km-1.
2. The descriptions on Figs. 5c and 5d in lines 320-322 seems not consistent with what was shown in figure. For example, the prevailed wind direction during spring and summer for TJ is southerly as shown in Fig. 5c, which is not the case stated by the text in lines 320-322, i.e. easterly wind is prevailed in TJ.

———————————————

---

## Author Comment (AC1) · 16 Dec 2017

Response to comments by referee 1

We would like to thank you for your comments and helpful suggestions. We revised our manuscript according to these comments and suggestions.

Specific comments:

This paper characterizes mixing layer height (MLH) over the major cities in the North China Plain based on the two-year surface observations. The relationship between MLH and regional air pollution is explored using concurrent PM, MLH, surface radiation, and meteorological parameters in the same cities. Overall, the paper is well written and the finding about the low MLH in southern Hebei is valuable to develop an efficient air pollution mitigation strategy in North China. I suggest the paper should be accepted by ACP after the authors address my comments below.

Comment 1:

It is not clear what is the difference between the MLH discussed here and the traditional defined planetary boundary layer height (PBLH). It would be interesting to see if the MLH obtained from surface can be inter-compared with PBLH from soundings like Guo J. et al. (2016).

Response 1:

Thank you for your helpful suggestion. Actually, we have already made comparisons between the MLH obtained from ceilometers and sounding data in Tang et al. (2016). The comparison results found that the ceilometers underestimate the MLH under conditions of neutral stratification caused by strong winds, whereas it overestimates MLH when sand-dust is crossing. When we excluded these two special weather conditions, the ceilometers observation results were fairly consistent with those retrieved from the sounding data. Besides, since the ceilometers can reflect the rainy conditions and the precipitation will influence the MLH retrieval, data for precipitation were also excluded. In our study, data rectifications were made at the BJ, SJZ, TJ and QHD stations. The criterion to exclude the data points with (a) precipitation, i.e. cloud base lower than 4000 m and the attenuated backscattering coefficient of at least $2 \times 10^{-6}$ $m^{-1}sr^{-1}$ within 0 m and the cloud base, (b) sandstorm, i.e. the ratio of $PM_{2.5}$ to $PM_{10}$ suddenly decreased to 30 % or lower and the $PM_{10}$ concentration was higher than 500 μg $m^{-3}$, and (c) strong winds, i.e. a sudden change in temperature and wind speed when cold fronts passed by (Muñoz and Undurraga, 2010; Tang et al., 2016; van der Kamp and McKendry, 2010). Relevant contents were modified in section 2.2 in the revised manuscript.

Comment 2:

L266, the authors attribute the lower summertime MLH in QHD to the higher frequency of sea breeze. However, the underlying physical mechanism is not fully explained. Intuitively, the active sea breezes should come with more unstable atmosphere over the land. Figure 5 about prevailing wind directions in different seasons is referred, but it is still unclear to me how this figure supports the hypothesis above. Some detailed discussions are needed to better describe the formation and

characteristics of the sea breeze in the coastal regions.

**Response 2:**

Thank you for your helpful suggestion. We are sorry for the unclear illustration about the impact of sea breezes. Here, we remade the monthly diurnal wind vectors and shown below in Fig.1. We can see that the sea breeze usually started at midday (approximately 11:00 LT) and prevailed during daytime at the QHD station in spring and summer (Fig. 1d). The sea breeze usually brings cold and stable air mass from the sea to the coastal region. Under the influence of the abrupt change of aerodynamic roughness and temperature between the land and sea surfaces, a thermal internal boundary layer (TIBL) will form in the coastal areas. Then the local mixing layer will be replaced by the TIBL. Under the influence of warm air on the land, the sea air advects downwind and warms, leading to a weak temperature difference between the air and the ground. In consequence, the TIBL warms less rapidly due to the decreased heat flux at the ground, and the rise rate reduced. Besides, since the TIBL deepens with distance downwind and usually can not extend all the way to the top of the intruding marine air, the remaining of the cool marine air above the TIBL will hinder the TIBL vertical development (Stull, 1988; Sicard et al., 2006). As a result, the MLH at the QHD station was lower than other stations from April to September. Since this south-southwesterly wind impacts enhanced in summer due to the weak synoptic systems, frequent occurrence of the TIBL resulted in the lowest MLH at the QHD station in summer. To better illustrate the sea breezes impacts, we also made relevant modifications in section 3.2.2 in the revised manuscript.

[Figure]

Fig. 1 Monthly variations of diurnal wind vectors at the BJ, SJZ, TJ and QHD stations from December 2013 to November 2014.

**Comment 3:**

L372, to overcome the lack of radio sounding in SJZ, how about directly using the reanalysis data? The quality of reanalysis can be evaluated by radio-sound at XT.

**Response 3:**

Thank you for your suggestion. We have made comparisons between reanalysis data and observation data at the Xingtai (XT) and Laoting (LT) stations, respectively. The reanalysis data were downloaded from the ECMWF website (http://apps.ecmwf.int/datasets/data/interim-full-mnth/levtype=pl/). As shown in Fig.2, there were large discrepancies between the two data sets. Meanwhile, the vertical resolution of reanalysis data was too low to calculate the wind shear profile. Therefore, the reanalysis data could not be used to describe the meteorological parameters variations in this study. Considering the absence of vertical meteorological observations in other stations, comparisons of wind speed between the XT and Shijiazhuang (SJZ) stations, as well as LT and Qinhuangdao (QHD) stations were also made with the reanalysis data (Fig. 3). Wind speed between the XT and SJZ stations, LT and QHD stations were highly correlated, respectively, which indicated that the wind speed in SJZ and QHD could be replaced by data in the XT and LT stations, respectively.

[Figure]

Fig. 2 Comparisons of seasonal wind speed profiles between the reanalysis and observation data at (a) the XT stations and (b) the LT stations.

[Figure]

Fig. 3 Comparisons of seasonal wind speed profiles between the (a) XT and SJZ stations and the (b) LT and QHD stations with reanalysis data.

**Comment 4:**

Section 4.1, could absorbing aerosols be another factor to explain the reason of the low MLH in southern Hebei? Observations have revealed that the ambient aerosols can become highly absorptive in the urban conditions in China [Peng J. et al., 2016, PNAS]. The strong solar absorption near the top of PBL can increase the atmospheric stability and convective inhibition energy [Wang Y. et al., 2013, AE; Li Z. et al., 2016, Rev. Geos.]. Those possible influences from the feedback of air pollution should be discussed and quantified if possible.

**Response 4:**

Thank you for your constructive suggestion very much. We have read your mentioned papers and some other relevant researches. Absorbing aerosols above the MLH can be another factor affecting the MLH, it give rise to an increasing temperature aloft but a decreasing temperature at the surface, which will enhance the strength of capping inversion and inhibit the convective ability (Peng et al., 2016; Wang et al., 2013b; Li et al., 2016). On the contrary, absorbing aerosols within the mixing layer could reduce the capping inversion intensity despite the reduction in the surface buoyancy flux and raise the MLH (Yu et al., 2002). Considering the higher concentrations of surface $PM_{2.5}$ in southern Hebei, absorbing aerosols could be likely to have some impacts on MLH development. However, the comprehensive influences from the feedback of absorbing aerosols above and below the MLH are hard to explain without sufficient knowledge of vertical variations in absorbing aerosols. Although the near-ground absorbing aerosol concentration (such as black carbon) has regional differences (Zhao et al., 2013), the absorbing aerosol column concentrations could be consistent (Gong

et al., 2017) with little difference in absorptive aerosol optical depths (AAOD). Besides, the mixed state and morphology of absorbing aerosols also dominate the absorption effects (Jacobson, 2001; Bond et al., 2013). Therefore, without sufficient observation data, it is hardly to discuss and quantify the possible influences from the feedback of air pollution on MLH development at present. Some elaborate experiments of vertical profiles and morphology need to be implemented in future studies. To compensate for this deficiency and let readers know the uncertainties, the relevant contents were modified in section 4.1 in the revised manuscript.

**Comment 5:**
L437, what makes the RH at SJZ is higher than that in BJ and TJ? SJZ is more inland than those two cities.

**Response 5:**
Thank you for your suggestion. As shown in Fig. 4, seasonal distributions of near-ground RH from December 2013 to November 2014 in the NCP were depicted below. It was obvious that the southern Hebei had higher RH than that in the northern NCP. The RH distribution was not only related to the distance from the sea, but also to the flow fields and synoptic systems. This might be resulted from the frequent passage of Siberian high in the northern NCP, especially in spring and winter. In spring, when frequent sand storm happens, it brings dry air mass to the northern NCP, thus the RH in northern NCP was far less than that in southern Hebei (Fig. 4a). Meanwhile, under the impact of Siberian High, frequent weak northwest flow from the Inner Mongolia will bring cold and dry air to the northern NCP in winter and autumn, and such north flow was too weak to reach southern Hebei (Su et al., 2004), which will lead to lower RH in the northern NCP (Fig. 4c and 4d). Besides, the higher RH in the southern Hebei could also be affected by the subtropical high (wet southeast flow) from the yellow sea.

[Figure]

Fig. 4 Distributions of seasonal averaged RH in the NCP from December 2013 to November 2014: (a) spring, (b) summer, (c) autumn and (d) winter.

**Comment 6:**
L432-445, some basics of new particle formation in urban condition should be thoroughly reviewed. Please refer to Zhang, R. 2010, Science and 2015, Rev. Chem.
**Response 6:**
Thank you for your helpful suggestion. We are sorry for our superficial cognition about the new particle formation and growth processes. We remade some figures to illustrate the annual means of RH and T distributions over north China (Fig. 5). The T value in the southern Hebei was similar to that in the northern NCP (Fig. 5a), which indicated an almost consistent temperature condition for atmospheric chemical reaction between these two areas (Seinfeld., 1998; Zhang et al., 2010; Zhang et al., 2015). However, differences existed in RH between southern Hebei and the northern NCP. The RH in the southern Hebei was always higher than that in the northern NCP (Fig. 5b). As our response to your comment 5, the Siberian and the subtropical high will be responsible for such RH distribution in the NCP region. Since the RH is a key factor for haze development, higher RH is beneficial to fine particles growth through hygroscopic growth process and heterogeneous reactions. Relevant contents were modified in section 4.2.1 in the revised manuscript.

[Figure]

Fig. 5 Distributions of annual means of (a) T and (b) RH over the NCP region from December 2013 to November 2014.

**Comment 7:**

Fig. 8. Define $V_c$ in the figure caption.

**Response 7:**

Thank you for your suggestion. We have already added the definition for $V_c$ in the figure caption of Fig. 9 in the revised manuscript.

**References:**

Bond, T., et al.: Bounding the role of black carbon in the climate system: a scientific assessment, J. Geophys. Res., 118, 1-173, doi:10.1002/jgrd.50171, 2013.

Gong, C., J. X., S. Wang, Y. Wang, T. Zhang: Anthropogenic aerosol optical and radiative properties in the typical urban/suburban regions in China, Atmos. Res., doi:10.1016/j.atmosres.2017.07.002, 2017.

Jacobson, M.: Strong radiative heating due to the mixing state of black carbon in atmospheric aerosols, Nature, 409,695-697, 2001.

Li, Z., et al.: Aeosol and monsoon climate interactions over Asia, Rev. Geophys., 54, 886-929, doi:10.1002/2015RG000500, 2016.

Muñoz, R. and A. Undurraga: Daytime Mixing layer over the Santiago Basin: Description of Two Years of Observations with a Lidar Ceilometer, J. Appl. Meteorol. Clim., 49(8), 1728-1741, doi:10.1175/2010jamc2347.1, 2010.

Peng, J., M. Hu, S. Guo, Z. Du, J. Zheng, D. Shang, M. L. Zamora, L. Zeng, M. Shao, Y. Wu, J. Zheng, Y. Wang, C. R. Glen, D. R. Collins, M. J. Molina, and R. Zhang: Markedly enhanced absorption and direct radiative forcing of black carbon under polluted urban environments, P. Natl. Acad. Sci. Usa., 113(4266-4271), doi:10.1073/pnas.1602310113, 2016.

Sicard, M., Pérez, C., Rocadenbosch, F., Baldasano, J.M. and García-Vizcaino, D.: 2006. Mixed-Layer Depth Determination in the Barcelona Coastal Area From Regular Lidar Measurements: Methods, Results and Limitations. Boundary-Layer Meteorology 119, 135-157.

Stull, R.B.: An Introduction to Boundary Layer Meteorology, Kluwer Academic

Publishers, Dordrecht, 1988.

Su F. Q., M. Z. Yang, J. H. Zhong and Z. G. Zhang: The effects of synoptic type on regional atmospheric contamination in North Chian, Res. Of Environ. Sci., 17(3), doi:10.13198/j.res.2004.03.18.sufq.006, 2004.

Seinfeld J. H. and S.N. Pandis: Atmospheric Chemistry and Physics: From Air Pollution to Climate Change, New York: John Wiley and Sons, 1998.

Tang, G., J. Zhang, X. Zhu, T. Song, C. Münkel, B. Hu, K. Schäfer, Z. Liu, J. Zhang, L. Wang, J. Xin, P. Suppan, and Y. Wang, Mixing layer height and its implications for air pollution over Beijing, China, Atmospheric Chemistry and Physics, 16, 2459-2475, doi:10.5194/acp-16-2459-2016, 2016.

Van der Kamp, D., and I. McKendry: Diurnal and Seasonal Trends in Convective Mixed-Layer Heights Estimated from Two Years of Continuous Ceilometer Observations in Vancouver, BC, Bound.-Lay. Meteorol., 137(3), 459-475, doi:10.1007/s10546-010-9535-7, 2010.

Wang, Y., M. L. Zamora, and R. Zhang: New Directions: Light absorbing aersols and their atmospheric impacts, Atmos. Environ., 81, 713-715, doi: 10.1016/j.atmosenv.2013.09.034, 2013b.

Wei J., G. Tang, X. Zhu, L. Wang, Z. Liu, M. Cheng, C. Münkel, X. Li and Y. Wang: Thermal internal boundary layer and its effects on air pollutants during summer in a coastal city in North China, Journal of Environmental Sciences, 1001-0742, doi:10.1016/j.jes.2017.11.006, 2017.

Yu, H., S. Liu, and R. Dickinson: Radiative effects of aerosols on the evolution of the atmospheric boundary layer, J. Geo. Res.: Atmos., 107, D12(4142), doi:10.1029/2001JD000754, 2002.

Zhao, P., F. Dong, Y. Yang, D. He, X. Zhao, W. Zhang, Q. Yao, H. Liu: Characteristics of carbonaceous aerosol in the region of Beijing, Tianjin, and Hebei, China, Atmos. Environ., 71, 389-398, doi: 10.1016/j.atmosenv.2013.02.010, 2013.

Zhang, R.: Getting to the Critical Nucleus of Aerosol Formation, Science, 328(5984), 1366-1367, doi: 10.1126/science.1189732, 2010.

Zhang, R., G. Hui, S. Guo, M. L. Zamora, Q. Ying, Y. Lin, W. Wang, M. Hu, and Y. Wang: Formation of Urban Fine Particulate Matter, Chem. Rev., 115, 3803-3855, doi: 10.1021/acs.chemrev.5b00067, 2015.

---

## Author Comment (AC2) · 16 Dec 2017

Response to comments by referee 2

We would like to thank you for your comments and helpful suggestions. We revised our manuscript according to these comments and suggestions.

Specific comments:

This study reveals the spatial variation of mixing layer height (MLH) over northern China plain (NCP) based on a two-year measurement at four primary cities with different geographic allocation across NCP. The authors attribute the different spatial pattern of MLH between southern Hebei and northern NCP to the distinct wind shear features between the two interested regions. The analysis on the long-term measurement of MLH in this study provides a meaningfully insight on the climatological features of boundary layer condition during the haze episodes over NCP. Also, the discussions about the associations of MLH and other meteorological factors with the near-ground particle pollution are sufficiently presented in this work. However, the following concerns should be addressed before publication.

Comment 1:

Considering the possible strong aerosol-radiation interaction because of the heavily pollution, the surface net radiation is supposed to be lower over the regions with more heavily pollution because of the strong scattering and/or absorbing of solar radiation by aerosols. However, in this study, though the near-ground $PM_{2.5}$ concentration over southern Hebei is 1.3 times higher than that of north China plain (NCP), there is no significant difference in the net radiation at Shijiazhuang (SJZ) located southern Hebei from at Beijing (BJ) located over NCP. One probable reason is because the aerosol optical depth (AOD) over the two sites was comparable, leading to comparable capacity reducing solar radiation. The authors may check the AOD data to obtain a convinced explanation for why the net radiation is spatial consistent, given the presence of aerosol-radiation interaction.

Response 1:

Thank you for your helpful suggestion. We have checked the AOD distribution in NCP as you suggested. The AOD data were retrieved with the dark target algorithm from the Moderate Resolution Imaging Spectra-radiometer (MODIS) aerosol products on board the NASA EOS (Earth observing system) Terra satellite. As shown in Fig. 1 below, the AOD in Shijiazhuang (SJZ) was 1.1and 1.0 times higher than that at the Beijing (BJ) and Tianjin (TJ) stations, respectively. Given the presence of aerosol-radiation interaction, the comparative amount of AOD could be one probable reason to explain the nearly consistent net radiation between the SJZ and BJ stations. In our revised manuscript, the net radiation analysis was replaced by gradient Richardson number ($Ri$) studies, and $Ri$ is a better index which can evaluate the turbulent stability from both of the perspective of thermal and mechanism forces. Then the low mixing layer height (MLH) in winter in southern Hebei was mainly resulted from the stable turbulent stratification (Fig.1). Relevant contents were modified in section 4.1 in the revised manuscript. Besides, we also discovered some new findings when the analysis of AOD was added in the discussion. Please refer to

comment 2.

[Figure]

Fig.1 Vertical profiles of (a, e) horizontal WS, (b, f) wind shear, (c, g) virtual potential temperature gradient and (d, h) percentage of $Ri>1$ at the BJ, XT and LT stations in summer (upper panel) and winter (lower panel).

**Comment 2:**
In addition to the difference in mixing layer height (MLH), how likely does the spatial variation in pollutant emissions contribute to the difference in the near-ground PM pollution between SJZ and BJ?

**Response 2:**
Thank you for your suggestion. Since the particle has direct emission sources and secondary sources, and the distribution of direct emissions cannot represent the total contribution of emissions to the particle concentration. The near-ground $PM_{2.5}$ concentration could represent the particle concentrations at the ground, but considering that the lifetime of particle is much longer than that of trace gases, the concentrations of particles are nearly uniform in the mixing layer because of the strong vertical mixing. Therefore, near-ground $PM_{2.5}$ concentrations cannot be used to evaluate the emissions influences between different regions if the mixing layer heights are different. AOD, which represent the aerosol column concentration, is a much better indicator for the emissions difference. As shown in Fig. 2, the major sites in southern Hebei (the SJZ, Handan (HD) and Xingtai (XT) stations) and northern NCP (the BJ, and TJ stations) were circled with white rectangles. The averaged AOD value at the southern Hebei stations was 1.2 times higher than the AOD at the northern NCP regions, while the near-ground $PM_{2.5}$ concentration in southern Hebei was 1.5 times higher than that in the northern NCP. If the difference of AOD

represents the emission discrepancy, the remaining differences of PM$_{2.5}$ concentration may be induced by the meteorology. In other words, except for the emission effect, the meteorological conditions also play an important role in pollutant contrast between these two areas. Relevant contents were also modified in section 4.2 in our revised manuscript.

[Figure]

Fig. 2 Distribution of AOD from December 2013 to November 2014 in the NCP. The PM$_{2.5}$ concentrations of the 13 observation sites were also marked beside each station. Major sites in the Northern NCP (BJ and TJ) and the Southern Hebei (SJZ, XT and HD) were circled by white rectangles.

**Comment 3:**
The authors attribute the spatial difference in wind shear over NCP during winter to the influence of front passing associated with the Siberian High (lines 403-405). Is the front also the dominant control of the relative humidity over NCP during winter? Is there any other reason leading to the discrepancy in relative humidity between the two regions in question?

**Response 3:**
The spatial difference in wind shear over the NCP in spring, autumn and winter was probably resulted from the more frequent weak cold air impact on the northern NCP region. When the cold air was brought by a high-pressure system, the cold front formed and enhanced the wind shear in BJ. But in summer, due to the northward lift and westward intrusion of the subtropical high on the NCP, the lessened effect of the weak cold air on northern NCP accompanied with strong solar radiation and turbulent activities will lead to less wind shear contrast in the vertical direction between southern Hebei and the northern NCP. Certainly, the front is also the dominant control of the RH over NCP. And higher RH in southern Hebei might be resulted from the frequent passage of Siberian high in the north NCP, especially in spring and winter. In spring, when frequent sand storm happens, it brings dry air mass to the northern NCP,

thus the RH in northern NCP was far less than that in southern Hebei (Fig. 3a). Meanwhile, under the impact of Siberian High, frequent weak northwest flow from the Inner Mongolia will bring cold and dry air to the northern NCP in winter and autumn, and such north flow was too weak to reach southern Hebei, which will lead to lower RH in the northern NCP (Fig. 3c and 3d). Besides, the higher RH in the southern Hebei could also be affected by the subtropical high (wet southeast flow from the yellow sea).

[Figure]

Fig. 3 Distributions of seasonal averaged RH in the NCP from December 2013 to November 2014: (a) spring, (b) summer, (c) autumn and (d) winter.

**Comment 4:**

Given that both Tianjin (TJ) and Qinhuangdao (QHD) are located at coastal region and suffering highly frequent sea breezes during summer (Fig. 5), why the MLH of TJ is much higher than the case in QHD, since the relatively low MLH in QHD is attributed by the authors to the intensive occurrence of sea breeze during summer (lines 265-266)?

**Response 4:**

Thank you for your suggestion and we are sorry for our unclear description. Actually, the MLH at the coastal region was affected by the thermal internal boundary layer (TIBL), not the sea breeze. When the cold air mass came with sea breeze and the top of the mixing layer was higher than the top of the air mass, the TIBL will form within the original mixing layer, interrupt the original mixing layer development and decrease the MLH. With distance inland, the top of the sea air mass will enhance and exceed the local MLH, if so, the TIBL will not form. Therefore, although the sea

breeze impact will extend further inland, the TIBL impact will only matters within a distance of about 10 km out to the sea (Stull, 1988). Since the QHD station was only 2 km away from the coastline and the distance of TJ station was about 50 km out to sea, the TIBL will not form in the TJ station. The MLH for TJ was as high as those inland sites (BJ and SJZ). The relevant contents were modified in section 3.2.2 in our revised manuscript

**Technical comments:**
**Comment 1:**
Fig. 7: the unit for the wind shear should be m s-1 km-1.
**Response 1:**

Since the wind shear $= \sqrt{\left(\frac{\Delta \bar{u}}{\Delta z}\right)^2 + \left(\frac{\Delta \bar{v}}{\Delta z}\right)^2}$ and the unit of wind speed and $\Delta$ z was m

s$^{-1}$ and m, respectively, the unit of wind shear was m s$^{-1}$m$^{-1}$.

**Comment 2:**
The descriptions on Figs. 5c and 5d in lines 320-322 seems not consistent with what was shown in figure. For example, the prevailed wind direction during spring and summer for TJ is southerly as shown in Fig. 5c, which is not the case stated by the text in lines 320-322, i.e. easterly wind is prevailed in TJ.
**Response 2:**
Thank you for your suggestion and we have already modified the relevant descriptions in section 3.2.2 in the revised manuscript.

**References:**
Stull, R.B.: An Introduction to Boundary Layer Meteorology, Kluwer AcademicPublishers, Dordrecht, 1988.

---

## Author Comment (AC3) · 16 Dec 2017

**Response to short comments**

We would like to thank you for your comments and helpful suggestions. We revised our manuscript according to these comments and suggestions.

**Specific comments:**

The climatology of MLH at four sites over NCP was investigated using long-term measurements. However, lots of statements in the manuscript and part of conclusions were not well supported. Thus, a major revision is suggested.

**Comment 1:**

LINE 214-215, the definitions of rainy, sandstorm and windy conditions should be given.

**Response 1:**

Thank you for your suggestion. The criterion to exclude the data points for special conditions with (a) precipitation, i.e. cloud base lower than 4000 m and the attenuated backscattering coefficient of at least $2 \times 10^{-6}$ $m^{-1}sr^{-1}$ within 0 m and the cloud base, (b) sandstorm, i.e. the ratio of $PM_{2.5}$ to $PM_{10}$ suddenly decreased to 30 % or lower and the $PM_{10}$ concentration was higher than 500 $\mu g$ $m^{-3}$, and (c) strong winds, i.e. a sudden change in temperature and wind speed when cold fronts passed by. We also modified the relevant contents in section 2.2 in the revised manuscript.

**Comment 2:**

LINE 317-318, "the TJ station was supposed to be an inland site", the TJ site is quite close to the Bohai sea, which should be considered as a coastal station.

**Response 2:**

Actually, the Tianjin (TJ) site was set in the courtyard of the Tianjin Meteorological Bureau, which was located south of the urban area (117.20° E, 39.13° N) with about 50 km away from the coast. While, the Qinhuangdao (QHD) station was set up in the Environmental Management College of China (119.57° E, 39.95° N) with only about 2 km away from the coastline. Therefore, the TJ site, by contrast, was supposed to be an inland site. Besides, the mixing layer height (MLH) at the coastal region was affected by the thermal internal boundary layer (TIBL), not the sea breeze. When the cold air mass came with sea breeze and the top of the mixing layer was higher than the top of the air mass, the TIBL will form within the origin mixing layer, interrupt the origin mixing layer development and decrease the MLH. With distance inland, the top of the sea air mass will enhance and exceed the local MLH, if so, the TIBL will not form. Since the TIBL impact will only matters within a distance of about 10 km out to the sea (Stull, 1988), although The TJ station is close to the sea, the MLH in TJ was not influenced by the TIBL. From another point of view, the definition of a coastal station should be the one that was affected by the TIBL.

**Comment 3:**

LINE 319-324, the definition of sea-breeze used in this study should be given. The sea-breeze cannot be identified merely by the near-surface wind speed and direction.

How to identify the sea-breeze from background wind? How to calculate the occurrence frequency of sea-breeze at TJ and QHD?

**Response 3:**

Thank you for your suggestion. The sea-land breeze was a local circulation, it happens when there is no large scale synoptic system pass by. In our study, we first exclude days with large-scale synoptic systems. Then according to the coastline orientation, if the southeast wind at the TJ station and south and southwest winds at the QHD station occurred at around 11:00 LT, and the northwest wind started to blow at around 20:00 LT, then this kind of circulation was supposed to be a sea-land circulation. The prevailed southeast wind at the TJ station and the south and southwest wind at the QHD station were regarded as sea breezes.

**Comment 4:**

LINE 326-335, more evidences should be given to support the statement that the movement of sea-breeze suppress the MLH at QHD site in summer. The TJ site also locates in the coastal regions, why the diurnal patterns and seasonal variations of MLH are quite different?

**Response 4:**

Thank you for your suggestion. Here, we remade the monthly diurnal wind vectors and shown below in Fig.1. We can see that the sea breeze usually started at midday (approximately 11:00 LT) and prevailed during daytime at the QHD station in spring and summer (Fig. 1d). The sea breeze usually brings cold and stable air mass from the sea to the coastal region. Under the influence of the abrupt change of aerodynamic roughness and temperature between the land and sea surfaces, a TIBL will form in the coastal areas. Then the local mixing layer will be replaced by the TIBL. Under the influence of warm air on the land, the sea air advects downwind and warms, leading to a weak temperature difference between the air and the ground. In consequence, the TIBL warms less rapidly due to the decreased heat flux at the ground, and the rise rate reduced. Besides, since the TIBL deepens with distance downwind and usually can not extend all the way to the top of the intruding marine air, the remaining of the cool marine air above the TIBL will hinder the TIBL vertical development (Stull, 1988; Sicard et al., 2006). As a result, the MLH at the QHD station was lower than other stations from April to September. Since this south-southwesterly wind impacts enhanced in summer due to the weak synoptic systems, frequent occurrence of the TIBL resulted in the lowest MLH at the QHD station in summer. Since the MLH at the coastal region was affected by the TIBL, not the sea breeze, and the TIBL impact will only matters within a distance of about 10 km out to the sea (Stull, 1988), the TIBL will not form in the TJ station. The MLH for TJ was as high as those inland sites (Beijing (BJ) and Shijiazhuang (SJZ)). The relevant contents were modified in section 3.2.2 in our revised manuscript

[Figure]

Fig. 1 Monthly variations of prevailed wind at the BJ, SJZ, TJ and QHD stations from December 2013 to November 2014.

**Comment 5:**

LINE 362-364, the buoyancy fluxes are determined by the surface sensible heat fluxes, not the net radiations. The statements here are inaccurate.

**Response 5:**

Thank you for your suggestion. The sensible heat fluxes data were not available, so we used net radiation for the analysis. Considering your suggestion, the net radiation analysis was replaced by gradient Richardson number ($Ri$) studies, and $Ri$ is an index which can evaluate the turbulent stability from both of the perspective of thermal and mechanism forces. Then the low MLH in southern Hebei was mainly resulted from the stable turbulent stratification (Fig.2). Relevant contents were modified in section 4.1 in the revised manuscript.

[Figure]

Fig.2 Vertical profiles of (a, e) horizontal WS, (b, f) wind shear, (c, g) virtual potential temperature gradient and (d, h) percentage of $Ri>1$ at the BJ, XT and LT stations in summer (upper panel) and winter (lower panel).

**Comment 6:**

LINE 371-375, before using the sounding data of XT as a replacement of SJZ, the data consistency must be examined and presented, since there are90 km between these two sites. At least, the general characteristics of MLH at SJZ at 08:00 and 20:00 LT should be well reflected by the sounding data at XT. The data consistency also should be check between the LT site and QHD site.

**Response 6:**

Thank you for your suggestion. Since we didn't have sounding data at the SJZ and QHD stations, we used the reanalysis data to do the examination instead. The reanalysis data were downloaded from the ECMWF website (http://apps.ecmwf.int/datasets/data/interim-full-mnth/levtype=pl/). Take the wind speed as an example, comparisons of the wind speed between the Xingtai (XT) and SJZ stations and the Laoting (LT) and QHD stations were shown in Fig.3. Wind speed between the XT and SJZ stations, LT and QHD stations were highly correlated, respectively, which indicated that the sounding data in the SJZ and QHD stations could be replaced by data in the XT and LT stations, respectively.

[Figure]

Fig. 3 Comparisons of seasonal wind speed profiles between the (a) XT and SJZ stations and the (b) LT and QHD stations with reanalysis data.

**Comment 7:**

As shown in Fig. 7, the profiles at XT are almost the same in different season and different moment, which is different from the profiles of other sites. The prevailing wind speed and direction are different in different season, why the profiles are almost the same? The error-bar of the profiles should also be given. In spring and summer, at 20:00 LT there are lots of fluctuations in the profiles at LT, why? Do the terrains play a role in the profiles in different regions?

**Response 7:**

Although the prevailing wind speed and direction at the XT station were different in different moment and season, vertical variation of each wind speed profiles changed slightly. Since the wind shear was defined as the degree of wind speed and direction variation between the upper layer and the lower layer

( wind shear $= \sqrt{\left(\frac{\Delta \bar{u}}{\Delta z}\right)^2 + \left(\frac{\Delta \bar{v}}{\Delta z}\right)^2}$ ), the almost consistent wind shear profiles in

different season and different moment indicated a relatively stable atmosphere stratification. Similarly, the stronger variation and higher value of wind shear in the vertical direction at the BJ station suggested unstable atmosphere stratification, which was probably due to the frequent occurrence of cold air mass passage. From Fig. 1 we can see that the sea breeze changed to land breeze at around 20:00 LT, thus the fluctuations in the profiles at LT could be attributed to the transitory stages of sea-land breeze alternation. Therefore, the terrains certainly play a role in the wind shear profiles in different regions. To further interpret the reasons for low MLH at the

southern Hebei, we added analysis of gradient Richardson number ($Ri$) profiles at the BJ, XT and LT stations in the revised manuscript. Since the comparison results at 08:00 LT and 20:00 LT made no difference, we combined the sounding profiles at 08:00 LT and 20:00 LT to make our paper concise and easier to be understood (Fig. 2). Then the low MLH in southern Hebei was mainly resulted from the stable turbulent stratification

**Comment 8:**
LINE 390-392, the authors merely presented the profiles at 20:00 LT, which cannot support the statement "during the whole night". More evidences should be given.
**Response 8:**
Thank you for your suggestion. In our revised manuscript, the meteorological profiles were averaged over 08:00 LT and 20:00 LT, and the wind shear and virtual potential temperature gradient profiles were compared between southern Hebei and northern NCP (Fig. 2). The wind shear in southern Hebei was lower than that in the southern NCP above 300 m, while the virtual potential temperature gradient was on the opposite, leading to a conclusion that the low MLH in southern Hebei was resulted from the stable turbulent stratification. In summer, this discrepancy was largely decreased and the MLHs were consistent between these two areas. The relevant contents were modified in section 4.1 in the revised manuscript.

**Comment 9:**
LINE 404-405, please give evidences to support the statement "the front usually does not reach southern Hebei".
**Comment 10:**
LINE 406-408, please give evidences to support the statement "the lessened effects of the front system and strong turbulent exchange will lead to less wind shear contrast in the vertical direction between southern Hebei and the northern NCP."
**Response 9 and 10:**
Thank you for your suggestion and we are sorry for our misrepresentation. Although haze evolution in the NCP area usually behave regional consistency, the pollution intensity various in different regions, which will be partially attributed to the impact of different position of weather system. The NCP region usually influenced by the continental high in the spring, autumn and winter in lower troposphere. When the high pressure is relatively weak, the northern and southern areas usually located in the front and south of the system, respectively. Thus, the weak cold and clean air may be partially responsible for the lighter pollution degree in the northern NCP areas (Su et al., 2004), meanwhile, the cold front resulted from the cold air flow over the northern NCP will enhanced the wind shear. In summer, due to the northward lift and westward intrusion of the subtropical high on the NCP, the lessened effect of the weak cold air on northern NCP accompanied with strong solar radiation and turbulent activities will lead to less wind shear contrast in the vertical direction between southern Hebei and the northern NCP. Based on this, we have made modifications in section 4.1 in our revised manuscript.

**Comment 11:**

LINE 410-419, the authors attribute the high PM concentration in SJZ to the low MLH. It is inaccurate, the different anthropogenic emissions of pollutants in SJZ and BJ should be considered.

**Response 11:**

Thank you for your suggestion. Since the particle has direct emission sources and secondary sources, and the distribution of direct emissions cannot represent the total contribution of emissions to the particle concentration. The near-ground $PM_{2.5}$ concentration could represent the particle concentrations at the ground, but considering that the lifetime of particle is much longer than that of trace gases, the concentrations of particles are nearly uniform in the mixing layer because of the strong vertical mixing. Therefore, near-ground $PM_{2.5}$ concentrations cannot be used to evaluate the emissions influences between different regions if the mixing layer heights are different. AOD, which represent the aerosol column concentration, is a much better indicator for the emissions difference. In the revised manuscript, we checked the AOD distribution in NCP to evaluate the emission effect. The AOD data were retrieved with the dark target algorithm from the Moderate Resolution Imaging Spectra-radiometer (MODIS) aerosol products on board the NASA EOS (Earth observing system) Terra satellite. As shown in Fig. 4, the averaged AOD value at the southern NCP (SJZ, Handan (HD) and Xingtai (XT)) stations was 1.2 times higher than the AOD at the northern NCP (the BJ, and TJ stations) regions, while the near-ground $PM_{2.5}$ concentration in southern Hebei was 1.5 times higher than that in the northern NCP. If the difference of AOD represents the emission discrepancy, the remaining differences of $PM_{2.5}$ concentration may be induced by the meteorology. In other words, except for the emission effect, the meteorological conditions also play an important role in pollutant contrast between these two areas. Relevant contents were also modified in section 4.2 in our revised manuscript.

[Figure]

Fig. 4 Distribution of AOD from December 2013 to November 2014 in the NCP. The PM$_{2.5}$ concentrations of the 13 observation sites were also marked beside each station. Major sites in the Northern NCP (BJ and TJ) and the Southern Hebei (SJZ, XT and HD) were circled by white rectangles.

**Comment 12:**
LINE 420-422, although the RH can affect the visibility, it cannot significantly affect the aerosol concentration. Is there any direct physical connections between the high RH conditions and high aerosol concentration?

**Response 12:**
The RH can not only affect the visibility, but also the aerosol concentrations. The direct physical mechanism is the fine particle's hygroscopic growth and the RH has positive correlation with fine particle's number and mass concentrations (Hu et al., 2006; Liu et al., 2011; Seinfeld et al., 1998).

**Comment 13:**
LINE 426-427, "temperature is the main factor in new particle formation," any evidences to support this statement in NCP.

**Response 13:**
Thank you for your suggestion and we are sorry for this inappropriate illustration. Actually, the temperature has impact on the particles physicochemical reaction rate, the particles' nucleation and other secondary transformation processes are most efficient in a relatively high temperature and RH. If the temperature was lower than the ideal value, the aerosol's secondary transformation processes will be less effective (Seinfeldet al., 1998).

**Comment 14:**
LINE 437-440, the RH in SJZ is higher than that in TJ (closer to sea), why?

**Response 14:**
Thank you for your suggestion. As shown in Fig. 5, seasonal distributions of near-ground RH from December 2013 to November 2014 in the NCP were depicted below. It was obvious that the southern Hebei had higher RH than that in the northern NCP. The RH distribution was not only related to the distance from the sea, but also to the flow fields and synoptic systems. This might be resulted from the frequent passage of Siberian high in the north NCP, especially in spring and winter. In spring, when frequent sand storm happens, it brings dry air mass to the northern NCP, thus the RH in northern NCP was far less than that in southern Hebei (Fig. 5a). Meanwhile, under the impact of Siberian High, frequent weak northwest flow from the Inner Mongolia will bring cold and dry air to the northern NCP in winter and autumn, and such north flow was too weak to reach southern Hebei (Su et al., 2004), which will lead to lower RH in the northern NCP (Fig. 5c and 5d). Besides, the higher RH in the southern Hebei could also be affected by the subtropical high (wet southeast flow from the yellow sea).

[Figure]

Fig. 5 Distributions of seasonal averaged RH in the NCP from December 2013 to November 2014: (a) spring, (b) summer, (c) autumn and (d) winter.

**Comment 15:**

Section 4.2.1, the authors attribute the higher PM in SJZ to new particle formation, which is quite complex and cannot be understood merely by the surface temperature and RH. And the direct emissions of pollutants should be considered.

**Response 15:**

Thank you for your suggestion. Since the particle has direct emission sources and secondary sources, and the distribution of direct emissions cannot represent the total contribution of emissions to the particle concentration. The near-ground $PM_{2.5}$ concentration could represent the particle concentrations at the ground, but considering that the lifetime of particle is much longer than that of trace gases, the concentrations of particles are nearly uniform in the mixing layer because of the strong vertical mixing. Therefore, near-ground $PM_{2.5}$ concentrations cannot be used to evaluate the emissions influences between different regions if the mixing layer heights are different. AOD, which represent the aerosol column concentration, is a much better indicator for the emissions difference. As shown in Fig. 4, the averaged AOD value at the southern NCP (SJZ, HD and XT) stations was 1.2 times higher than the AOD at the northern NCP (BJ and TJ) regions, while the near-ground $PM_{2.5}$ concentration in southern Hebei was 1.5 times higher than that in the northern NCP. If the difference of AOD represents the emission discrepancy, the remaining differences of $PM_{2.5}$ concentration may be induced by the meteorology. In other words, except for the emission effect, the meteorological conditions also play an important role in

pollutant contrast between these two areas. The lower MLH combined with higher RH and weaker diffusion ability contributed a lot to heavier haze in the southern Hebei and the meteorological contrast between these two areas will accounted for 60% in the near-ground $PM_{2.5}$ concentration difference. Relevant contents were also modified in section 4.2 in our revised manuscript.

**Comment 16:**
LINE 470-473, "it was considered reasonable to regard the sounding data of WS as a climatological constant", during a day, the WS within ML would change due to the momentum exchanges between the ML and free troposphere. The WS cannot be considered as a constant. As illustrated in Fig. S2, there are differences in profiles at 08:00 and 20:00 LT. The error-bar of wind speed should be given.

**Response 16:**
Thank you for your suggestion, we are sorry for this inaccurate expression. The wind speed in our study was supposed to be a climatological feature, not a climate constant. And the wind speeds at 08:00 LT and 20:00 LT were used to calculate the ventilation coefficient approximately. Although it will be better to include the sounding data at noon, this is the best choice at present due to the acquired data confined. Relevant contents were supplemented in the conclusion section to explain the uncertainties of our study.

References:
Hu M., S. Liu, Z. J. Wu, J. Zhang, Y. L. Zhao, W. Birgit, and W. Alfred: Effects of high temperature, high relative humidity and rain process on particle size distributions in the summer of Beijing, Environ. Sci., 27(11), 2006.

Liu Z. R., Y. Sun, L. Li and Y. S. Wang: Particle mass concentrations and size distribution during and after the Beijing Olympic Games, Environ. Sci., 32(4), doi:10.13227/j.hjkx.2011.04.015, 2011.

Sicard, M., Pérez, C., Rocadenbosch, F., Baldasano, J.M. and García-Vizcaino, D.: 2006. Mixed-Layer Depth Determination in the Barcelona Coastal Area From Regular Lidar Measurements: Methods, Results and Limitations. Boundary-Layer Meteorology 119, 135-157.

Stull, R.B.: An Introduction to Boundary Layer Meteorology, Kluwer Academic Publishers, Dordrecht, 1988.

Su F. Q., M. Z. Yang, J. H. Zhong and Z. G. Zhang: The effects of synoptic type on regional atmospheric contamination in North Chian, Res. Of Environ. Sci., 17(3), doi:10.13198/j.res.2004.03.18.sufq.006, 2004.

Seinfeld J. H. and S.N. Pandis: Atmospheric Chemistry and Physics: From Air Pollution to Climate Change, New York: John Wiley and Sons, 1998.

---

## Author Response (AR3)

**Response to comments by referee 1**

We would like to thank you for your comments and helpful suggestions. We revised our manuscript according to these comments and suggestions.

**Specific comments:**

This paper characterizes mixing layer height (MLH) over the major cities in the North China Plain based on the two-year surface observations. The relationship between MLH and regional air pollution is explored using concurrent PM, MLH, surface radiation, and meteorological parameters in the same cities. Overall, the paper is well written and the finding about the low MLH in southern Hebei is valuable to develop an efficient air pollution mitigation strategy in North China. I suggest the paper should be accepted by ACP after the authors address my comments below.

**Comment 1:**

It is not clear what is the difference between the MLH discussed here and the traditional defined planetary boundary layer height (PBLH). It would be interesting to see if the MLH obtained from surface can be inter-compared with PBLH from soundings like Guo J. et al. (2016).

**Response 1:**

Thank you for your helpful suggestion. Actually, we have already made comparisons between the MLH obtained from ceilometers and sounding data in Tang et al. (2016). The comparison results found that the ceilometers underestimate the MLH under conditions of neutral stratification caused by strong winds, whereas it overestimates MLH when sand-dust is crossing. When we excluded these two special weather conditions, the ceilometers observation results were fairly consistent with those retrieved from the sounding data. In addition, since the ceilometers can reflect the rainy conditions and precipitation will influence the MLH retrieval, data for precipitation were also excluded. In our study, data rectifications were made at the BJ, SJZ, TJ and QHD stations. The criterion to exclude these data points is as follows: (a) precipitation, i.e., a cloud base lower than 4000 m and the attenuated backscattering coefficient of at least $2 \times 10^{-6}$ $m^{-1}sr^{-1}$ within 0 m and the cloud base, (b) sandstorm, i.e., the ratio of $PM_{2.5}$ to $PM_{10}$ suddenly decreased to 30 % or lower and the $PM_{10}$ concentration was higher than 500 μg $m^{-3}$, and (c) strong winds, i.e., a sudden change in temperature and wind speed when cold fronts passed by (Muñoz and Undurraga, 2010; Tang et al., 2016; Kamp and McKendry, 2010). Relevant contents were modified in section 2.2 in the revised manuscript.

**Comment 2:**

L266, the authors attribute the lower summertime MLH in QHD to the higher frequency of sea breeze. However, the underlying physical mechanism is not fully explained. Intuitively, the active sea breezes should come with more unstable atmosphere over the land. Figure 5 about prevailing wind directions in different seasons is referred, but it is still unclear to me how this figure supports the hypothesis above. Some detailed discussions are needed to better describe the formation and

characteristics of the sea breeze in the coastal regions.

**Response 2:**

Thank you for your helpful suggestion. We are sorry for the unclear illustration about the impact of sea breezes. Here, we re-created the monthly diurnal wind vectors as shown below in Fig.1. We can see that the sea breeze usually started at midday (approximately 11:00 LT) and prevailed during daytime at the QHD station in spring and summer (Fig. 1d). The sea breeze usually brings a cold and stable air mass from the sea to the coastal region. Under the influence of the abrupt change of aerodynamic roughness and temperature between the land and sea surfaces, a thermal internal boundary layer (TIBL) will form in the coastal areas. Then, the local mixing layer will be replaced by the TIBL. Under the influence of warm air on land, the sea air advects downwind and warms, leading to a weak temperature difference between the air and the ground. In consequence, the TIBL warms less rapidly due to the decreased heat flux at the ground, and the rise rate is reduced. In addition, since the TIBL deepens with distance downwind and usually can not extend all the way to the top of the intruding marine air, the remaining cool marine air above the TIBL will hinder the TIBL vertical development (Stull, 1988; Sicard et al., 2006). As a result, the MLH at the QHD station was lower than other stations from April to September. Since the south-southwesterly wind impacts were enhanced in summer due to the weak synoptic systems, a frequent occurrence of the TIBL resulted in the lowest MLH at the QHD station in summer. To better illustrate the sea breeze impacts, we also made relevant modifications in section 4.1 in the revised manuscript.

[Figure]

Fig. 1 Monthly variations of diurnal wind vectors at the BJ, SJZ, TJ and QHD stations from December 2013 to November 2014.

**Comment 3:**

L372, to overcome the lack of radio sounding in SJZ, how about directly using the reanalysis data? The quality of reanalysis can be evaluated by radio-sound at XT.

**Response 3:**

Thank you for your suggestion. We have made comparisons between reanalysis data and observation data at the Xingtai (XT) and Laoting (LT) stations, respectively. The reanalysis data were downloaded from the ECMWF website (http://apps.ecmwf.int/datasets/data/interim-full-mnth/levtype=pl/). As shown in Fig.2, there were large discrepancies between the two data sets. Meanwhile, the vertical resolution of the reanalysis data was too low to calculate the wind shear profile. Therefore, the reanalysis data could not be used to describe the meteorological parameter variations in this study. Considering the absence of vertical meteorological observations in other stations, comparisons of wind speed between the XT and Shijiazhuang (SJZ) stations, as well as LT and Qinhuangdao (QHD) stations were also made with the reanalysis data (Fig. 3). The wind speed between the XT and SJZ stations, and the LT and QHD stations were highly correlated, respectively, which indicated that the wind speeds in SJZ and QHD could be replaced by data in the XT and LT stations, respectively.

[Figure]

Fig. 2 Comparisons of seasonal wind speed profiles between the reanalysis and observation data at (a) the XT stations and (b) the LT stations.

[Figure]

Fig. 3 Comparisons of seasonal wind speed profiles between the (a) XT and SJZ stations and the (b) LT and QHD stations with reanalysis data.

**Comment 4:**

Section 4.1, could absorbing aerosols be another factor to explain the reason of the low MLH in southern Hebei? Observations have revealed that the ambient aerosols can become highly absorptive in the urban conditions in China [Peng J. et al., 2016, PNAS]. The strong solar absorption near the top of PBL can increase the atmospheric stability and convective inhibition energy [Wang Y. et al., 2013, AE; Li Z. et al., 2016, Rev. Geos.]. Those possible influences from the feedback of air pollution should be discussed and quantified if possible.

**Response 4:**

Thank you for your constructive suggestion very much. We have read your mentioned papers and some other relevant research. Absorbing aerosols above the MLH can be another factor affecting the MLH because it gives rise to an increasing temperature aloft but a decreasing temperature at the surface, which will enhance the strength of capping inversion and inhibit the convective ability (Peng et al., 2016; Wang et al., 2013; Li et al., 2016). In contrast, absorbing aerosols within the mixing layer could reduce the capping inversion intensity despite the reduction in the surface buoyancy flux and raise the MLH (Yu et al., 2002). Considering the higher concentrations of surface $PM_{2.5}$ in southern Hebei, absorbing aerosols could likely have some impacts on MLH development. However, the comprehensive influences from the feedback of absorbing aerosols above and below the MLH are difficult to explain without sufficient knowledge of the vertical variations in absorbing aerosols. Although the near-ground absorbing aerosol concentration (such as black carbon) has regional differences (Zhao et al., 2013), the absorbing aerosol column concentrations could be

consistent (Gong et al., 2017) with little difference in absorptive aerosol optical depths (AAOD). In addition, the mixed state and morphology of absorbing aerosols dominate the absorption effects (Jacobson, 2001; Bond et al., 2013). Therefore, without sufficient observation data, it is difficult to discuss and quantify the possible influences from the feedback of air pollution on the MLH development at present. Some elaborate experiments of vertical profiles and morphology need to be implemented in future studies. To compensate for this deficiency and inform readers of the uncertainties, the relevant contents were modified in section 4.2 in the revised manuscript.

**Comment 5:**
L437, what makes the RH at SJZ is higher than that in BJ and TJ? SJZ is more inland than those two cities.

**Response 5:**
Thank you for your suggestion. As shown in Fig. 4, seasonal distributions of near-ground RH from December 2013 to November 2014 in the NCP were depicted below. It was obvious that southern Hebei had higher RH than that in the northern NCP. The RH distribution was not only related to the distance from the sea but also to the flow fields and synoptic systems. This might result from the frequent passage of the Siberian High in the northern NCP, especially in spring and winter. In spring, when frequent sand storms occur, a dry air mass is brought to the northern NCP; thus the RH in the northern NCP was far less than that in southern Hebei (Fig. 4a). Meanwhile, under the impact of the Siberian High, a frequent weak northwest flow from Inner Mongolia will bring cold and dry air to the northern NCP in winter and autumn, and such north flow was too weak to reach southern Hebei (Su et al., 2004), which will lead to a lower RH in the northern NCP (Fig. 4c and 4d). In addition, the higher RH in southern Hebei could also be affected by the subtropical high (wet southeast flow) from the Yellow Sea.

[Figure]

Fig. 4 Distributions of seasonal averaged RH in the NCP from December 2013 to November 2014: (a) spring, (b) summer, (c) autumn and (d) winter.

**Comment 6:**

L432-445, some basics of new particle formation in urban condition should be thoroughly reviewed. Please refer to Zhang, R. 2010, Science and 2015, Rev. Chem.

**Response 6:**

Thank you for your helpful suggestion. We apologize for our superficial understanding of the new particle formation and growth processes. We re-created some figures to illustrate the annual means of RH and T distributions over north China (Fig. 5). The T value in southern Hebei was similar to that in the northern NCP (Fig. 5a), which indicated an almost consistent temperature condition for an atmospheric chemical reaction between these two areas (Seinfeld J. and S. Pandis, 1998; Zhang et al., 2010; Zhang et al., 2015). However, differences existed in RH between southern Hebei and the northern NCP. The RH in southern Hebei was always higher than that in the northern NCP (Fig. 5b). As mentioned in our response to your comment 5, the Siberian and the subtropical high will be responsible for this RH distribution in the NCP region. Since the RH is a key factor for haze development, higher RH is beneficial to fine particle growth through the hygroscopic growth process and heterogeneous reaction. Relevant contents were modified in section 4.3.1 in the revised manuscript.

[Figure]

Fig. 5 Distributions of annual means of (a) T and (b) RH over the NCP region from December 2013 to November 2014.

**Comment 7:**
Fig. 8. Define $V_c$ in the figure caption.

**Response 7:**
Thank you for your suggestion. We have already added the definition for $V_c$ in the figure caption of Fig. 9 in the revised manuscript.

**Response to comments by referee 2**

We would like to thank you for your comments and helpful suggestions. We revised our manuscript according to these comments and suggestions.

**Specific comments:**
This study reveals the spatial variation of mixing layer height (MLH) over northern China plain (NCP) based on a two-year measurement at four primary cities with different geographic allocation across NCP. The authors attribute the different spatial pattern of MLH between southern Hebei and northern NCP to the distinct wind shear features between the two interested regions. The analysis on the long-term measurement of MLH in this study provides a meaningfully insight on the climatological features of boundary layer condition during the haze episodes over NCP. Also, the discussions about the associations of MLH and other meteorological factors with the near-ground particle pollution are sufficiently presented in this work. However, the following concerns should be addressed before publication.

**Comment 1:**
Considering the possible strong aerosol-radiation interaction because of the heavily pollution, the surface net radiation is supposed to be lower over the regions with more heavily pollution because of the strong scattering and/or absorbing of solar radiation by aerosols. However, in this study, though the near-ground $PM_{2.5}$ concentration over southern Hebei is 1.3 times higher than that of north China plain (NCP), there is no significant difference in the net radiation at Shijiazhuang (SJZ) located southern Hebei from at Beijing (BJ) located over NCP. One probable reason is because the aerosol optical depth (AOD) over the two sites was comparable, leading to comparable capacity reducing solar radiation. The authors may check the AOD data to obtain a convinced explanation for why the net radiation is spatial consistent, given the presence of aerosol-radiation interaction.

**Response 1:**
Thank you for your helpful suggestion. We have checked the AOD distribution in the NCP as you suggested. The AOD data were retrieved with the dark target algorithm from the Moderate Resolution Imaging Spectra-radiometer (MODIS) aerosol products on board the NASA EOS (Earth observing system) Terra satellite. As shown in Fig. 1 below, the AOD in Shijiazhuang (SJZ) was 1.1 and 1.0 times higher than that at the Beijing (BJ) and Tianjin (TJ) stations, respectively. Given the presence of aerosol-radiation interaction, the comparative amount of AOD could be one probable reason to explain the nearly consistent net radiation between the SJZ and BJ stations. In our revised manuscript, the net radiation analysis was replaced by gradient Richardson number (*Ri*) studies, and *Ri* is a better index that can evaluate the atmospheric stability from both the perspectives of thermal and mechanism forces. Then, the low mixing layer height (MLH) in winter in southern Hebei mainly resulted from the stable turbulent stratification (using summer and winter as examples) (Fig.1). Relevant contents were modified in section 4.2 in the revised manuscript. In addition, we discovered some new findings when the AOD analysis was added in the

discussion. Please refer to comment 2.

[Figure]

Fig.1 Vertical profiles of (a, e) the horizontal WS, (b, f) the shear term, (c, g) the buoyancy term and (d, h) the percentage of $Ri>1$ at the BJ, XT and LT stations in summer (upper panel) and winter (lower panel).

**Comment 2:**
In addition to the difference in mixing layer height (MLH), how likely does the spatial variation in pollutant emissions contribute to the difference in the near-ground PM pollution between SJZ and BJ?

**Response 2:**
Thank you for your suggestion. Since the particle has direct emission sources and secondary sources, and the distribution of direct emissions cannot represent the total contribution of emissions to the particle concentration. The near-ground $PM_{2.5}$ concentration could represent the particle concentrations at the ground, but considering that the particle lifetime is much longer than that of trace gases, the particle concentrations are nearly uniform in the mixing layer because of the strong vertical mixing. Therefore, near-ground $PM_{2.5}$ concentrations cannot be used to evaluate the emissions influences between different regions if the mixing layer heights are different. AOD, which represents the aerosol column concentration, is a much better indicator for the emissions difference. As shown in Fig. 2, the major sites in southern Hebei (the SJZ, Handan (HD) and Xingtai (XT) stations) and the northern NCP (the BJ and TJ stations) were enclosed with white rectangles. The average AOD value at the southern Hebei stations was 1.2 times higher than the AOD at the northern NCP regions, while the near-ground $PM_{2.5}$ concentration in southern Hebei was 1.5 times higher than that in the northern NCP. If the AOD difference represents the emission discrepancy, the remaining differences of $PM_{2.5}$ concentration may be

induced by the meteorology. In other words, except for the emission effect, the meteorological conditions also play an important role in pollutant contrast between these two areas. Relevant contents were also modified in section 4.3 in our revised manuscript.

[Figure]

Fig. 2 AOD distribution from December 2013 to November 2014 in the NCP. The PM$_{2.5}$ concentrations of the 13 observation sites were also marked beside each station. Major sites in the northern NCP (BJ and TJ) and southern Hebei (SJZ, XT and HD) were enclosed by white rectangles.

**Comment 3:**
The authors attribute the spatial difference in wind shear over NCP during winter to the influence of front passing associated with the Siberian High (lines 403-405). Is the front also the dominant control of the relative humidity over NCP during winter? Is there any other reason leading to the discrepancy in relative humidity between the two regions in question?

**Response 3:**
The spatial difference in wind shear over the NCP in spring, autumn and winter probably resulted from the more frequent weak cold air impact on the northern NCP region. When the cold air was brought by a high-pressure system, the cold front formed and enhanced the wind shear in BJ. However, in summer, due to the northward lift and westward intrusion of the subtropical high on the NCP, the diminished effect of the weak cold air on the northern NCP accompanied with strong solar radiation and turbulent activities will lead to less wind shear contrast in the vertical direction between southern Hebei and the northern NCP. Certainly, the front is also the dominant control of the RH over the NCP. In addition, higher RH in southern Hebei might result from the frequent passage of the Siberian High in the north NCP, especially in spring and winter. In spring, when frequent sand storms occur, a dry air mass is brought to the northern NCP; thus, the RH in the northern NCP was far less

than that in southern Hebei (Fig. 3a). Meanwhile, under the impact of the Siberian High, a frequent weak northwest flow from Inner Mongolia will bring cold and dry air to the northern NCP in winter and autumn, and such north flow was usually too weak to reach southern Hebei (Su et al., 2004), which will lead to lower RH in the northern NCP (Figs. 3c and 3d). In addition, the higher RH in southern Hebei could also be affected by the subtropical high (wet southeast flow from the Yellow Sea).

[Figure]

Fig. 3 Distributions of seasonal averaged RH in the NCP from December 2013 to November 2014: (a) spring, (b) summer, (c) autumn and (d) winter.

**Comment 4:**
Given that both Tianjin (TJ) and Qinhuangdao (QHD) are located at coastal region and suffering highly frequent sea breezes during summer (Fig. 5), why the MLH of TJ is much higher than the case in QHD, since the relatively low MLH in QHD is attributed by the authors to the intensive occurrence of sea breeze during summer (lines 265-266)?

**Response 4:**
Thank you for your suggestion, and we apologize for our unclear description. Actually, the MLH at the coastal region was affected by the thermal internal boundary layer (TIBL), not the sea breeze. When the cold air mass came with the sea breeze and the top of the mixing layer was higher than the top of the air mass, the TIBL will form within the original mixing layer, interrupting the original mixing layer development and decreasing the MLH. With distance inland, the top of the sea air mass will enhance and exceed the local MLH; if so, the TIBL will not form, and the TIBL

impact will be impaired with distance inland (Stull, 1988). Since the QHD station was only 2 km away from the coastline and the distance of the TJ station was approximately 50 km out to sea, the TIBL will not form in the TJ station. The MLH variation for TJ was the same with those inland sites (BJ and SJZ). The relevant contents were modified in section 4.1 in our revised manuscript.

**Technical comments:**
**Comment 1:**
Fig. 7: the unit for the wind shear should be m s-1 km-1.
**Response 1:**

Since the wind shear $= \sqrt{\left(\frac{\Delta \bar{u}}{\Delta z}\right)^2 + \left(\frac{\Delta \bar{v}}{\Delta z}\right)^2}$ and the unit of wind speed and $\Delta z$ was m

s$^{-1}$ and m, respectively, the unit of wind shear was m s$^{-1}$m$^{-1}$. The study of wind shear was replaced by the study of shear term $\left(\left(\frac{\Delta \bar{u}}{\Delta z}\right)^2 + \left(\frac{\Delta \bar{v}}{\Delta z}\right)^2\right)$ in our revised manuscript. And to be consistent with the unit of buoyancy term, the unit of shear term was s$^{-2}$.

**Comment 2:**
The descriptions on Figs. 5c and 5d in lines 320-322 seems not consistent with what was shown in figure. For example, the prevailed wind direction during spring and summer for TJ is southerly as shown in Fig. 5c, which is not the case stated by the text in lines 320-322, i.e. easterly wind is prevailed in TJ.
**Response 2:**
Thank you for your suggestion, and we have already modified the relevant descriptions in section 4.1 in the revised manuscript.

**Response 2:**
Actually, the Tianjin (TJ) site was set in the courtyard of the Tianjin Meteorological Bureau, which was located south of the urban area (117.20°E, 39.13°N) with approximately 50 km away from the coast. The Qinhuangdao (QHD) station was set up in the Environmental Management College of China (119.57°E, 39.95°N) with only approximately 2 km away from the coastline. Therefore, the TJ site, by contrast, was supposed to be an inland site. In addition, the mixing layer height (MLH) at the coastal region was affected by the thermal internal boundary layer (TIBL), not the sea breeze. When the cold air mass came with the sea breeze and the top of the mixing layer was higher than the top of the air mass, the TIBL will form within the origin mixing layer, interrupt the origin mixing layer development and decrease the MLH. With distance inland, the top of the sea air mass will enhance and exceed the local MLH; if so, the TIBL will not form, and the TIBL impact will be impaired with distance inland (Stull, 1988). Since the QHD station was only 2 km away from the coastline and the distance of the TJ station was approximately 50 km out to sea, the TIBL will not form in the TJ station. The MLH variation for TJ was the same with those inland sites. The relevant contents were modified in section 4.1 in our revised manuscript. From another point of view, the definition of a coastal station should be the one that was affected by the TIBL.

**Comment 3:**

LINE 319-324, the definition of sea-breeze used in this study should be given. The sea-breeze cannot be identified merely by the near-surface wind speed and direction. How to identify the sea-breeze from background wind? How to calculate the occurrence frequency of sea-breeze at TJ and QHD?

**Response 3:**

Thank you for your suggestion. The sea-land breeze was a local circulation; it occurs when there is no large scale synoptic system that passes. In our study, we first exclude days with large-scale synoptic systems. Then according to the coastline orientation, if the southeast wind at the TJ station and south and southwest winds at the QHD station occurred at approximately 11:00 LT, and the northwest wind started to blow at approximately 20:00 LT, then this type of circulation was supposed to be a sea-land circulation. The prevailing southeast wind at the TJ station and the south and southwest wind at the QHD station were regarded as sea breezes.

**Comment 4:**

LINE 326-335, more evidences should be given to support the statement that the movement of sea-breeze suppress the MLH at QHD site in summer. The TJ site also locates in the coastal regions, why the diurnal patterns and seasonal variations of MLH are quite different?

**Response 4:**

Thank you for your suggestion. Here, we re-created the monthly diurnal wind vectors as shown below in Fig.1. We can see that the sea breeze usually started at midday (approximately 11:00 LT) and prevailed during daytime at the QHD station in spring and summer (Fig. 1d). The sea breeze usually brings a cold and stable air mass from the sea to the coastal region. Under the influence of the abrupt change of aerodynamic roughness and temperature between the land and sea surfaces, a TIBL will form in the coastal areas. Then, the local mixing layer will be replaced by the TIBL. Under the influence of warm air on land, the sea air advects downwind and warms, leading to a weak temperature difference between the air and the ground. In consequence, the TIBL warms less rapidly due to the decreased heat flux at the ground, and the rise rate is reduced. In addition, since the TIBL deepens with distance downwind and usually can not extend all the way to the top of the intruding marine air, the remaining cool marine air above the TIBL will hinder the TIBL vertical development (Stull, 1988; Sicard et al., 2006). As a result, the MLH at the QHD station was lower than other stations from April to September. Since the south-southwesterly wind impacts are enhanced in summer due to the weak synoptic systems, a frequent occurrence of the TIBL resulted in the lowest MLH at the QHD station in summer. As a result, the MLH at the coastal region was affected by the TIBL, not the sea breeze, and the TIBL impact will be impaired with distance inland (Stull, 1988). Since the TJ station was approximately 50 km out to sea, the TIBL will not extend inland so far, and the MLH in TJ had no influence from the TIBL, leading to the same MLH variation with those inland sites (Beijing (BJ) and Shijiazhuang (SJZ)). The relevant contents were modified in section 4.1 in our revised manuscript.

[Figure]

Fig. 1 Monthly variations of prevailing wind at the BJ, SJZ, TJ and QHD stations from December 2013 to November 2014.

**Comment 5:**

LINE 362-364, the buoyancy fluxes are determined by the surface sensible heat fluxes, not the net radiations. The statements here are inaccurate.

**Response 5:**

Thank you for your suggestion. The sensible heat fluxes data were not available, so we used net radiation for the analysis. Considering your suggestion, the net radiation analysis was replaced by gradient Richardson number ($Ri$) studies, and $Ri$ is an index that can evaluate the turbulent stability from both the perspectives of thermal and mechanism forces. Then, the low MLH in southern Hebei mainly resulted from stable turbulent stratification (Fig.2). Relevant contents were modified in section 4.2 in the revised manuscript.

[Figure]

Fig.2 Vertical profiles of (a, e) the horizontal WS, (b, f) the shear term, (c, g) the buoyancy term and (d, h) the percentage of $Ri>1$ at the BJ, XT and LT stations in summer (upper panel) and winter (lower panel).

**Comment 6:**

LINE 371-375, before using the sounding data of XT as a replacement of SJZ, the data consistency must be examined and presented, since there are90 km between these two sites. At least, the general characteristics of MLH at SJZ at 08:00 and 20:00 LT should be well reflected by the sounding data at XT. The data consistency also should be check between the LT site and QHD site.

**Response 6:**

Thank you for your suggestion. Since we did not have sounding data at the SJZ and QHD stations, we used the reanalysis data to perform the examination instead. The reanalysis data were downloaded from the ECMWF website (http://apps.ecmwf.int/datasets/data/interim-full-mnth/levtype=pl/). Using the wind speed as an example, comparisons of the wind speed between the Xingtai (XT) and SJZ stations and the Laoting (LT) and QHD stations are shown in Fig. 3. The wind speed between the XT and SJZ stations, and the LT and QHD stations were highly correlated, respectively, which indicated that the sounding data in the SJZ and QHD stations could be replaced by data in the XT and LT stations, respectively.

[Figure]

Fig. 3 Comparisons of seasonal wind speed profiles between the (a) XT and SJZ stations and the (b) LT and QHD stations with reanalysis data.

**Comment 7:**
As shown in Fig. 7, the profiles at XT are almost the same in different season and different moment, which is different from the profiles of other sites. The prevailing wind speed and direction are different in different season, why the profiles are almost the same? The error-bar of the profiles should also be given. In spring and summer, at 20:00 LT there are lots of fluctuations in the profiles at LT, why? Do the terrains play a role in the profiles in different regions?

**Response 7:**
Although the prevailing wind speed and direction at the XT station were different in different moments and seasons, the vertical variation of each wind speed profile changed slightly. Since the wind shear was defined as the degree of wind speed and direction variation between the upper layer and the lower layer (wind shear =

$$\sqrt{\left(\frac{\Delta \bar{u}}{\Delta z}\right)^2 + \left(\frac{\Delta \bar{v}}{\Delta z}\right)^2}$$ ), the almost consistent wind shear profiles in different seasons and

different moments indicated a relatively stable atmospheric stratification. Similarly, the stronger variation and higher value of wind shear in the vertical direction at the BJ station suggested a relatively unstable atmospheric stratification, which was probably due to the frequent passage of cold air masses. Fig. 1 shows that the sea breeze changed to land breeze at approximately 20:00 LT; thus, the fluctuations in the profiles at LT could be attributed to the transitory stages of sea-land breeze alternation. Therefore, the terrains certainly play a role in the wind shear profiles in different regions. To further interpret the reasons for low MLH at southern Hebei, we included

an analysis of gradient Richardson number (*Ri*) profiles at the BJ, XT and LT stations in the revised manuscript and the wind shear study was replaced by the study of shear term ($\left(\frac{\Delta \bar{u}}{\Delta z}\right)^2 + \left(\frac{\Delta \bar{v}}{\Delta z}\right)^2$). Since the comparison results at 08:00 LT and 20:00 LT made no difference, we combined the sounding profiles at 08:00 LT and 20:00 LT to make our paper concise and easily understood (Fig. 2). Then, the low MLH in southern Hebei mainly resulted from stable turbulent stratification

**Comment 8:**
LINE 390-392, the authors merely presented the profiles at 20:00 LT, which cannot support the statement "during the whole night". More evidences should be given.
**Response 8:**
Thank you for your suggestion. In our revised manuscript, the meteorological profiles were averaged over 08:00 LT and 20:00 LT, and the shear term and buoyancy term profiles were compared between southern Hebei and the northern NCP (Fig. 2). The wind shear term in southern Hebei was lower than that in the northern NCP within 0-1200 m in spring, autumn and winter, while the buoyancy term was on the opposite, leading to a conclusion that the low MLH in southern Hebei resulted from stable turbulent stratification. In summer, this discrepancy was largely decreased and the MLHs were consistent between these two areas. The relevant contents were modified in section 4.2 in the revised manuscript.

**Comment 9:**
LINE 404-405, please give evidences to support the statement "the front usually does not reach southern Hebei".
**Comment 10:**
LINE 406-408, please give evidences to support the statement "the lessened effects of the front system and strong turbulent exchange will lead to less wind shear contrast in the vertical direction between southern Hebei and the northern NCP."
**Response 9 and 10:**
Thank you for your suggestion and we are sorry for our misrepresentation. Although haze evolution in the NCP area is usually regionally consistent, the pollution intensity varies in different regions, which will be partially attributed to the impact of different positions of weather systems. The NCP region is usually influenced by the continental high in the spring, autumn and winter in the lower troposphere. When the high pressure is relatively weak, the northern and southern areas are usually located in front and to the south of the system, respectively. Thus, the weak cold and clean air may be partially responsible for the lighter pollution degree in the northern NCP areas (Su et al., 2004). Meanwhile, the cold front caused by the cold air flow over the northern NCP will enhance the shear term. In summer, due to the northward lift and westward intrusion of the subtropical high on the NCP, the diminished effect of the weak cold air on the northern NCP accompanied with strong solar radiation and turbulent activities will lead to less shear term contrast in the vertical direction between southern Hebei and the northern NCP. Based on this, we have made

modifications in section 4.2 in our revised manuscript.

**Comment 11:**
LINE 410-419, the authors attribute the high PM concentration in SJZ to the low MLH. It is inaccurate, the different anthropogenic emissions of pollutants in SJZ and BJ should be considered.

**Response 11:**
Thank you for your suggestion. Since the particle has direct emission sources and secondary sources, the direct emissions distribution cannot represent the total emissions contribution to the particle concentration. The near-ground $PM_{2.5}$ concentration could represent the particle concentrations at the ground, but considering that the lifetime of a particle is much longer than that of trace gases, the particle concentrations are nearly uniform in the mixing layer because of the strong vertical mixing. Therefore, near-ground $PM_{2.5}$ concentrations cannot be used to evaluate the emissions influences between different regions if the mixing layer heights are different. AOD, which represents the aerosol column concentration, is a much better indicator for the emissions difference. In the revised manuscript, we checked the AOD distribution in the NCP to evaluate the emissions effect. The AOD data were retrieved with the dark target algorithm from the Moderate Resolution Imaging Spectra-radiometer (MODIS) aerosol products on board the NASA EOS (Earth observing system) Terra satellite. As shown in Fig. 4, the averaged AOD value at southern Hebei (SJZ, Handan (HD) and Xingtai (XT)) was 1.2 times higher than the AOD at the northern NCP (BJ and TJ) region, while the near-ground $PM_{2.5}$ concentration in southern Hebei was 1.5 times higher than that in the northern NCP. If the difference of AOD represents the emissions discrepancy, the remaining differences of the $PM_{2.5}$ concentration may be induced by the meteorology. In other words, except for the emissions effect, the meteorological conditions also play an important role in pollutant contrast between these two areas. Relevant contents were also modified in section 4.3 in our revised manuscript.

[Figure]

Fig. 4 Distribution of AOD from December 2013 to November 2014 in the NCP. The PM$_{2.5}$ concentrations of the 13 observation sites were also marked beside each station. Major sites in the northern NCP (BJ and TJ) and southern Hebei (SJZ, XT and HD) are enclosed by white rectangles.

**Comment 12:**
LINE 420-422, although the RH can affect the visibility, it cannot significantly affect the aerosol concentration. Is there any direct physical connections between the high RH conditions and high aerosol concentration?
**Response 12:**
The RH can not only affect the visibility but also the aerosol concentrations. The direct physical mechanism is the fine particle's hygroscopic growth and the RH has a positive correlation with the fine particle's number and mass concentrations (Hu et al., 2006; Liu et al., 2011; Seinfeld et al., 1998).

**Comment 13:**
LINE 426-427, "temperature is the main factor in new particle formation," any evidences to support this statement in NCP.
**Response 13:**
Thank you for your suggestion, and we apologize for this inappropriate illustration. Actually, the temperature has impact on the particles physicochemical reaction rate. The particles' nucleation and other secondary transformation processes are most efficient in a relatively high temperature and RH. If the temperature was lower than the ideal value, the aerosol's secondary transformation processes would be less effective (Seinfeld et al., 1998).

**Comment 14:**
LINE 437-440, the RH in SJZ is higher than that in TJ (closer to sea), why?
**Response 14:**
Thank you for your suggestion. The seasonal distributions of near-ground RH from December 2013 to November 2014 in the NCP are depicted in Fig. 5. It is clear that southern Hebei had higher RH values than the northern NCP. The RH distribution was not only related to the distance from the sea but also to the flow fields and synoptic systems. This might resulted from the frequent passage of the Siberian High in the northern NCP, especially in spring and winter. In spring, when frequent sand storms occur, a dry air mass is brought to the northern NCP; thus, the RH in the northern NCP was far less than that in southern Hebei (Fig. 5a). Meanwhile, under the impact of the Siberian High, a frequent weak northwest flow from Inner Mongolia will bring cold and dry air to the northern NCP in winter and autumn, and the north flow was too weak to reach southern Hebei (Su et al., 2004), which will lead to a lower RH in the northern NCP (Fig. 5c and 5d). Additionally, the higher RH in southern Hebei could also be affected by the subtropical high in summer (wet southeast flow from the Yellow Sea) (Fig. 5b).

[Figure]

Fig. 5 Distributions of seasonal averaged RH in the NCP from December 2013 to November 2014: (a) spring, (b) summer, (c) autumn and (d) winter.

**Comment 15:**

Section 4.2.1, the authors attribute the higher PM in SJZ to new particle formation, which is quite complex and cannot be understood merely by the surface temperature and RH. And the direct emissions of pollutants should be considered.

**Response 15:**

Thank you for your suggestion. Since the particle has direct emissions sources and secondary sources, the distribution of direct emissions cannot represent the total contribution of emissions to the particle concentration. The near-ground $PM_{2.5}$ concentration could represent the particle concentrations at the ground, but considering that the particle lifetime is much longer than that of trace gases, the particle concentrations are nearly uniform in the mixing layer because of strong vertical mixing. Therefore, near-ground $PM_{2.5}$ concentrations cannot be used to evaluate the emissions influences between different regions if the mixing layer heights are different. AOD, which represents the aerosol column concentration, is a much better indicator for the emissions difference. As shown in Fig. 4, the averaged AOD value at southern Hebei (SJZ, HD and XT) was 1.2 times higher than the AOD at the northern NCP (BJ and TJ) region, while the near-ground $PM_{2.5}$ concentration in southern Hebei was 1.5 times higher than that in the northern NCP. If the difference of AOD represents the emissions discrepancy, the remaining differences of the $PM_{2.5}$ concentration may be induced by the meteorology. In other words, except for the

emissions effect, the meteorological conditions also play an important role in pollutant contrast between these two areas. The lower MLH combined with higher RH and weaker wind speed contributed to the heavier haze in southern Hebei. Relevant contents were also modified in section 4.3 in our revised manuscript.

**Comment 16:**
LINE 470-473, "it was considered reasonable to regard the sounding data of WS as a climatological constant", during a day, the WS within ML would change due to the momentum exchanges between the ML and free troposphere. The WS cannot be considered as a constant. As illustrated in Fig. S2, there are differences in profiles at 08:00 and 20:00 LT. The error-bar of wind speed should be given.

**Response 16:**
Thank you for your suggestion, and we apologize for this inaccurate expression. The wind speed in our study was supposed to be a climatological feature, not a climate constant. Additionally, the wind speeds at 08:00 LT and 20:00 LT were used to approximately calculate the ventilation coefficient. Although it will be better to include the sounding data at noon, this is the best choice at present due to the confined acquired data. Relevant contents were supplemented in the conclusion section to explain the uncertainties of our study.

**Response to comments by referee 3**
**Specific comments:**
The spatial and seasonal characteristics of mixing layer height (MLH) over northern China plain (NCP) were revealed by the authors using variety of measurements, primarily focusing on northern NCP and southern Hebei. The authors attempt to explain the different feature of MLH development between the two interested regions by examining observed wind shear, buoyancy and turbulent stability. In addition, this study pointed out that the MLH plays a key role in forming the heavy near-ground particular mater (PM) pollution besides emissions. The paper is well organized and the reasoning for MLH spatial variations and its association with air pollution is comprehensively discussed. I recommend to publish this paper in ACP journal after addressing following minor issues.

**Comment 1:**
For Fig. 6: What are the reasons for the difference in buoyancy term profiles between the site XT and BJ during winter, as shown in Fig. 6g? In addition, the absolute values of buoyancy term seem larger than that of shear term; does this mean the buoyancy term rather than the shear term is the dominant component in determining turbulent energy?

**Response 1:**
Thank you for your suggestion. As we described in section 2.4, the Gradient Richard

number ($Ri$) is the ratio of buoyancy term ($\frac{g}{\bar{\theta}}\frac{\Delta\bar{\theta}}{\Delta z}$) and shear term ($\left(\frac{\Delta\bar{u}}{\Delta z}\right)^2 + \left(\frac{\Delta\bar{v}}{\Delta z}\right)^2$).

For the static instability, the buoyancy term is usually negative, and the buoyancy force will suppress the turbulent development; for the neutral stratification, the buoyancy term is usually zero; and for the static stability stratification, the buoyancy is usually positive, which will promote the turbulent development. While the shear term usually has positive value and attribute to the mechanical turbulence. In our study, the averaged buoyancy term is positive and larger than the shear term, leading to the Ri larger than 1, this indicated that the mechanical production rate can not balance the turbulent kinetic energy's consumption by buoyancy. Therefore, from a statistical point of view, the atmospheric turbulence is stable on the BJ, XT and LT stations. Turbulent energy is affected by various factors, except for the buoyancy term and the mechanical product term, there are also the turbulent transport contribution and the dissipation terms. Through analysis of the $Ri$ value is just for a simplify and effective evaluation. Although the averaged result exhibited a more significant effect of buoyancy term than the shear term, there are many different occasions that the $Ri$ is less than 1, and the shear term may play a dominant role.

As we mentioned in line 407-410, the higher buoyancy term in XT may be resulted from the warm advection from the Loess Plateau. Since the warm advection usually develops from southwest to the northeast and results in strong thermal inversion above the NCP plain, the warm advection will has stronger impact at the XT station than the BJ station. Thus, this will lead to a higher buoyancy term contribution in XT.

**Comment 2:**

Line 381-382: It appears that the profiles were averaged over only two time points (i.e. 8:00 am and 08:00 pm), right? Can the average of only two time points represent the entire day MLH features which are most significant during noon time?

**Response 2:**

Thank you for your suggestion. Yes, the profiles were averaged over only two time points (i.e. 8:00 am and 08:00 pm). Analysis of these two time points could explain the lower MLH in XT at 8:00 am and 08:00 pm, but could not exactly illustrate the entire day MLH features. Although we can not better explain the day MLH features with these limited data, our study still provide a fundamental knowledge about the reasons for MLH contrast between northern NCP and southern Hebei. We can also make a prediction about these parameters features during noon time: it is known that the MLH development is mainly affected by the solar radiation during daytime and such radiation is almost consistent on the NCP plain, since the XT has weaker turbulent develop condition (i.e., weaker mechanical force and stronger buoyancy inhibition) than the BJ station, the MLH development at the XT station might be weaker than the BJ station. Such limitation in our study is illustrated in the final paragraph of the conclusion section.

**Comment 3:**

Line 444-447: In addition to the emission discrepancy and different meteorological factors (like MLH), the aerosol-radiation interactions are a potential candidate to explain different PM pollution between the two interested regions. Therefore, it is better to say 60% is the upper bound of contribution due to meteorological factors.

**Response 3:**

Thank you for your suggestion. Consider your comment, it is certainly reasonable to say 60 % is the upper bound of contribution due to meteorological factors. Therefore, the relevant contents were modified in section 4.3 in our revised manuscript.

**Comment 4:**

Line 474: a recent ref should also be cited here:

Wang G, et al. (2016) Persistent sulfate formation from London Fog to Chinese haze. Proceedings of the National Academy of Sciences 113(48):13630-13635.

**Response 4:**

Thank you for your suggestion. The paper that you mentioned has been cited in our revised manuscript.

**Comment 5:**

Line 409: "enhance" should be "enhanced" or "enhancement of".

**Response 5:**

Thank you for your suggestion. We have already modified the relevant content in the revised manuscript.

[revised manuscript text omitted]